# Assembling custom side chains on proteoglycans to interrogate their function in living cells

Wenshuang Wang[1], Naihan Han[1,2], Yingying Xu[1], Yunxue Zhao[3], Liran Shi[1], Jorge Filmus[4] & Fuchuan Li [1✉]

Proteoglycans (PGs) are composed of a core protein and one or more chains of glycosa-minoglycans (GAGs). The highly heterogeneous GAG chains play an irreplaceable role in the functions of PGs. However, the lack of an approach to control the exact structure of GAG chains conjugated to PGs tremendously hinders functional studies of PGs. Herein, by using glypican-3 as a model, we establish an aldehyde tag-based approach to assemble PGs with specific GAG chains on the surface of living cells. We show that the engineered glypican-3 can regulate Wnt and Hedgehog signaling like the wild type. Furthermore, we also present a method for studying the interaction of PGs with their target glycoproteins by combining the assembly of PGs carrying specific GAG chains with metabolic glycan labeling, and most importantly, we obtain evidence of GPC3 directly interacting with Frizzled. In conclusion, this study provides a very useful platform for structural and functional studies of PGs with specific GAG chains.

[1] National Glycoengineering Research Center and Shandong Provincial Key Laboratory of Carbohydrate Chemistry and Glycobiology, Shandong University, Qingdao, China. [2] Shandong Police College, Jinan, China. [3] Department of Pharmacology, School of Basic Medical Sciences, Shandong University, Jinan, China. [4] Sunnybrook Health Science Centre, University of Toronto, Toronto, Ontario, Canada. ✉email: fuchuanli@sdu.edu.cn

Proteoglycans (PGs) are complex biomacromolecules composed of a core protein covalently linked to one or more glycosaminoglycan (GAG) chains. PGs are ubiquitously found on the cell surface and in the extracellular matrix, and they participate in various biological events, such as cell adhesion and migration[1,2], cell signaling[1,3–8], cytokinesis[9], embryonic development[10–12], inflammation[13,14], and cancer progression[15]. Cell surface PGs may regulate cell signaling by interacting with various growth factors and morphogens such as fibroblast growth factor, vascular endothelial growth factor, Wnt, and Hedgehog[16]. The GAG chains of PGs play key roles in these interactions, although the core proteins are also involved in some cases.

GAG chains are negatively charged linear heteropolysaccharides composed of repeating disaccharide units. Based on the monosaccharide residues, the type of glycosidic bonds, and the number and position of the sulfate groups, GAGs can be classified into four main classes: hyaluronic acid (HA), chondroitin sulfate (CS)/dermatan sulfate (DS), heparin (Hep)/heparan sulfate (HS), and keratan sulfate (KS). Disaccharides of HA, CS/DS, and Hep/HS are composed of hexosamine (N-acetyl-galactosamine or N-acetyl-glucosamine) and uronic acid (D-glucuronic acid or L-iduronic acid), whereas the disaccharides of KS are composed of N-acetyl-glucosamine and galactose. Notably, HA is the only GAG that is not covalently attached to a core protein and is synthesized at the plasma membrane. In contrast to HA, all other GAGs are synthesized in the Golgi apparatus and are highly modified by specific enzymes including various sulfotransferases and the glucuronyl C-5 epimerase. A large number of studies have shown that the structures of GAGs depend on the cell type, and that these structures display specific biological functions.

Unlike nucleic acids and proteins, GAGs are synthesized in a nontemple-dependent manner and, therefore, GAG chains with specific structures cannot be generated by genetic manipulation. Currently, the function of GAG side chains is studied by mutating insertion sites in the core proteins, which results in the complete elimination of the GAGs. However, this approach does not allow to identify the role of specific GAG chain structures[17]. Recently, two studies reported cell-based methods to produce and display distinct GAGs with a broad repertoire of modifications by using precise gene editing[18,19]. Unfortunately, this method can only change the global structures of the GAG chains expressed by cells, but it cannot be used to control the structures of the GAG chains of specific PGs.

Formylglycine (fGly) is a cotranslational modification of cysteine (Cys) that can be used for site-specific protein conjugation[20,21]. A fGly-generating enzyme (FGE) can specifically recognize the short consensus sequence, CXPXR, where X represents any amino acid, except proline, and oxidize the Cys residue in this sequence to an unusual aldehyde-bearing fGly[22]. Aldehyde-tagged proteins can be easily modified with aminooxy-, hydrazine- or hydroxylamine-containing reagents at room temperature under weak acidic (pH 4–6) or even physiological conditions[22]. Active but rare aldehyde tags can be introduced in desired locations within any protein of interest by inserting the CXPXR sequence through site-directed mutagenesis. In this work, we use this approach to establish a method to insert specific GAG chains into an aldehyde-tagged PG core protein expressed on the surface of living cells. In addition, we combine this methodology with metabolic glycan editing to identify a glycoprotein that directly interacts with the tagged PG.

## Results

### Generation of hydrazide-labeled GAG oligosaccharides linked to AhGPC3.
Glypican-3 (GPC3) is one of the six members of the glypican family. Glypicans are HSPGs that are bound to the exocytoplasmic surface of the plasma membrane via a glycosyl-phosphatidylinositol anchor[23]. GPC3 displays two HS side chains that are linked to amino acid residues $Ser^{495}$ and $Ser^{509}$ via a tetrasaccharide linkage (GlcAβ1-3Galβ1-3Galβ1-4Xylβ1-O-Ser). These HS chains have been shown to play key roles in various physiological and pathological processes by mediating the interaction of GPC3 with growth factors/morphogens and their cognate receptors[13,24]. In this study, GPC3 was used as a model to develop our strategy for attaching specific GAG chains to PG core proteins located at the cell surface.

Although the aldehyde-tag technology has been used for the site-specific conjugation of various molecules, including GalNAc and N-galactosylated glycans, to proteins of interest, it has not been exploited for the modification of PG core proteins with GAG side chains. To investigate whether highly negative-charged GAG chains can be site-specifically conjugated to an aldehyde-tagged PG core protein, two glycanation site-including sequences ($E^{494}S^{495}G^{496}D^{497}C^{498}G^{499}$ and $G^{508}S^{509}G^{510}D^{511}G^{512}M^{513}$) of GPC3 were separately or jointly mutated to generate the consensus sequence LCTPSR. This sequence can be specifically recognized by hFGE to convert Cys to the aldehyde-bearing residue fGly (Fig. 1a)[22]. Thus, an aldehyde tag was introduced at the glycanation sites $Ser^{495}$ and $Ser^{509}$ of GPC3, respectively. Three mutants were then generated based on the location of the mutations: GPC3-S495C and GPC3-S509C with a single aldehyde tag at $Cys^{495}$ or $Cys^{509}$, respectively, and GPC3-O bearing aldehyde tags at both of these sites. In addition, to avoid the glycanation of the remaining sites, $Ser^{509}$ in GPC3-S495C and $Ser^{459}$ in GPC3-S509C were individually mutated to Ala. To confirm the conversion of Cys-to-fGly, the lysate of cells co-transfected with human FGE (hFGE) and the GPC3 mutant vector was treated with Biotin-LC-Hydrazide, and the avidin-binding capacity of the mutants was analyzed by a binding assay (Supplementary Note 1 and Supplementary Fig. 1). Furthermore, nanoscale liquid chromatography coupled to tandem mass spectrometry (NanoLC-MS/MS) analysis of GPC3-O showed that the dual conversion efficiency of $Cys^{495}$ and $Cys^{509}$ to fGly residues was 77.7% (Supplementary Note 1 and Supplementary Fig. 2). To conjugate specific GAG chains to the aldehyde-tagged GPC3 (AhGPC3) mutants, various size-defined GAG oligosaccharides were prepared by digestion with the corresponding GAG lyases and labeled with a hydrazide-tagged adipic dihydrazide at the reducing end (Supplementary Note 2 and Supplementary Fig. 3). The generation of hydrazide-labeled oligosaccharides (HaGAG) was further confirmed by high-performance liquid chromatography, electrospray ionization mass spectrometry, and $^{1}$H nuclear magnetic resonance (Supplementary Note 2 and Supplementary Figures 4–7). The labeling efficiencies of the various HaGAG oligosaccharides were estimated to be more than 30%.

To investigate whether the HaGAG oligosaccharides can be inserted into the AhGPC3 core proteins, GPC3-S495C and GPC3-O mutants were co-transfected with an hFGE expression vector into HEK 293T cells, which do not express endogenous GPC3. The cell lysates of the transfected cells were then incubated with hydrazide-labeled HS (HaHS) oligosaccharides in labeling buffer. Western blot (WB) analysis showed that compared to the unglycanated GPC3ΔGAG, GPC3-S495C (Fig. 1b), and GPC3-O (Fig. 1c) display a series of narrow bands with gradually increasing molecular mass as the size of the oligosaccharides increases, indicating that the HaHS oligosaccharides could be inserted into the core protein of AhGPC3. Based on the relative intensity of the bands, we estimate that ~55–70% of AhGPC3-S495C incorporated a size-defined HaHS (Fig. 1b and Supplementary Table 2). The reaction of GPC3-O with HaHS generated two bands in addition to the band corresponding to the core protein, indicating that the core protein with two aldehyde tags could be modified with either one or two HS oligosaccharide

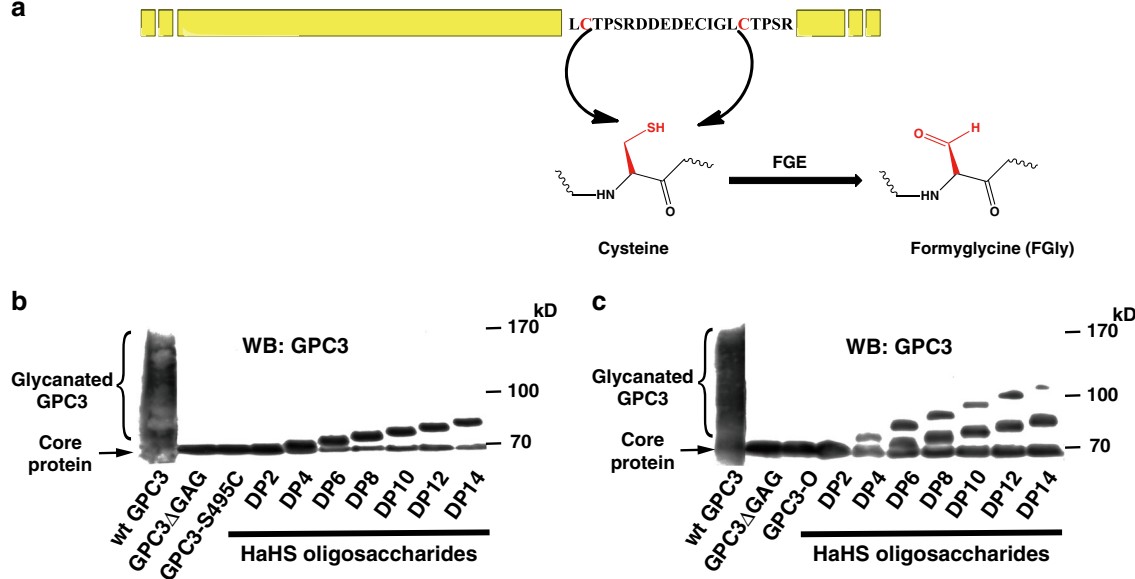

**Fig. 1 Specific oligosaccharides can be inserted into the GPC3 core protein. a** Schematic diagram of the methodology used to introduce the aldehyde-bearing residue fGly into the GPC3 core proteins present in cell lysates of transfected 293T cells. **b, c** Various HaHS oligosaccharides of the indicated sizes were conjugated to GPC3-S495C with a single aldehyde-tag (**b**) and to GPC3-O with two aldehyde tags (**c**). The levels of modified GPC3 core proteins were assessed by WB analysis with a GPC3 antibody. wt GPC3, wild-type GPC3. All experiments were repeated three times with similar results. Source data are provided in a Source Data file.

chains. The proportion of core proteins that were bound to one or two GAG chains represented ~40–50% or 15–40% of the total amount of GPC3 core protein, respectively (Fig. 1c and Supplementary Table 2), suggesting that the core protein is more easily conjugated with just one GAG chain. Notably, the efficiency of conjugation decreased with the size of the oligosaccharides (Fig. 1c and Supplementary Table 2). This inverse correlation is particularly obvious for the generation of the GPC3 with two GAG chains (Fig. 1c, Supplementary Fig. 10, and Supplementary Table 2). Overall, the results clearly show that after introduction of a hydrazide group at the reducing end, GAG oligosaccharides can be efficiently linked to the aldehyde-tagged core protein of GPC3.

**Insertion of hydrazide-labeled GAG oligosaccharides into AhGPC3 core proteins on the surface of living cells.** To insert HaHS oligosaccharide chains to GPC3-O expressed on the cell surface, AhGPC3-O-transfected 293T cells were incubated with an HaHS oligosaccharide mixture (3 µg/ml) or DP14 oligosaccharides (1 µM) under optimized conditions (pH 6.0, 37 °C for 4 h) (Supplementary Note 4 and Supplementary Fig. 9), and the corresponding cell lysates were analyzed by WB. As shown in Fig. 2b, similar to the lysate from cells transfected with wild-type GPC3, the lysate of cells transfected with the mutant AhGPC3-O and treated with the HaHS oligosaccharide mixture displays a smear. This smear could be remarkably intensified by concentrating the negatively charged PGs in the cell lysate using DEAE Sepharose. This result indicates that the HaHS oligosaccharides were successfully inserted into the AhGPC3-O present on the surface of living cells. Moreover, as found in Fig. 1c, two narrow bands corresponding to one and two chain-bound GPC3, were detected in the lysate of cells treated with the size-defined HaHS tetradecsaccharide (DP14) and these two bands could also be intensified by concentrating the PGs using DEAE Sepharose (Fig. 2b). Interestingly, compared to the results from the conjugation reaction in cell lysates (Fig. 1c), the proportion of unconjugated core protein in the living cells is relatively

high. We speculated that this could be due to the presence of unreacted AhGPC3-O inside the cell, where the HaHS oligosaccharides could not reach. To test this possibility, we used phosphatidylinositol-specific phospholipase C to release GPC3 from the cell membrane for WB analysis. As shown in Fig. 2b, line 7, the ratio of glycanated/non-glycanated GPC3 significantly increased, indicating that the conjugation reaction on living cell surface was as efficient in the living cell surface as in the cell lysate. In addition, the proportion of GPC3 that incorporated two HS chains is significantly increasead in the experiment performed with living cells (Fig. 2b), compared with that in cell lysates (Fig. 1c), although the reason for this difference remains to be investigated.

Next, the assembly of glycanated GPC3 on the cell surface was verified by immunofluorescence and by a fluorescence resonance energy transfer (FRET) assay. For the immunofluorescence experiment, AhGPC3-O-transfected cells were incubated with biotinylated HaGAG (bioHaGAG) oligosaccharides (Supplementary Notes 2 and 3, and Supplementary Figs. 3 and 8), and then the side chains and core protein of the engineered GPC3 on the cell surface were stained with tetramethylrhodamine (TRITC)-conjugated streptavidin and an anti-GPC3 monoclonal antibody, respectively. As shown in Fig. 2c, both stainings colocalized on the cell surface. Further quantitative analysis by flow cytometry showed that the fluorescence intensity of AhGPC3-O-transfected cells incubated with bioHaHS oligosaccharides was much stronger than those of negative controls (Fig. 2d). Next, FRET between the fluorescein isothiocyanate (FITC)-stained core protein and the TRITC-stained side chain in the same engineered GPC3 molecule was investigated (Fig. 3a). As shown in Fig. 3b, the excitation of FITC fluorescence (green) from the core protein induced a FRET signal (white) from the HS oligosaccharides (red) when AhGPC3-O on the cell surface was modified by bioHaHS. The FRET efficiency was ~38%. This result indicates that the GPC3 core protein and the HS chains are very close to each other. Taken together, these results further confirm the successful assembly of the engineered GPC3 on the surface of living cells.

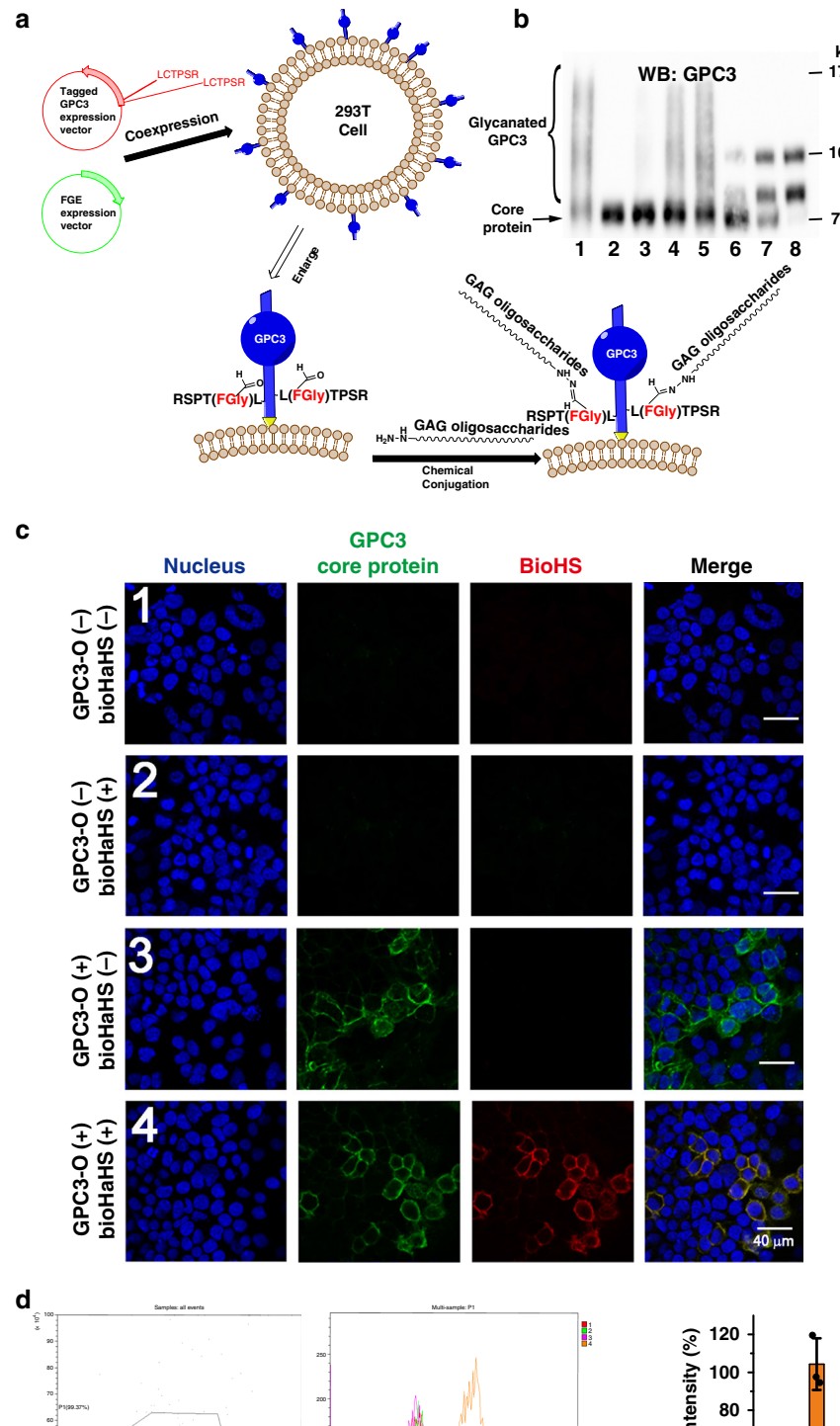

**Functional analysis of GPC3 attached to specific GAG chains.** It is well established that the function of PGs is highly dependent on the type of GAG chains attached to the core proteins. In the case of GPC3, we have previously demonstrated that the GAG chains are required for its optimal activity in the promotion of Wnt signaling[24], and in the inhibition of Hh signaling[25]. We decided therefore to investigate whether the engineered GPC3 could play a similar role as the wild-type GPC3 in Wnt and Hh signaling. The effect of the engineered GPC3 with specific GAG chains on the Wnt signaling pathway was investigated by performing a luciferase reporter assay with 293T cells, in which luciferase expression was driven by a Wnt-responsive promoter.

**Fig. 2 GAG oligosaccharides can be attached to GPC3-O in living cells. a** Schematic diagram of the attachment of HS oligosaccharides to the modified GPC3 core protein on the surface of living cells. **b** WB analysis of the attachment of HS to the GPC3 core proteins. Cells were transfected with the mutated GPC3-O core protein or vector control (pCMV6XL4) and hFGE vectors, and were then incubated with HaHS oligosaccharides. To better show that the HS side chains were efficiently attached to GPC3, the PGs were concentrated by DEAE Sepharose chromatography or directly released from cell surface by phosphatidylinositol-specific phospholipase C, as described under "Methods". 1, wild-type GPC3; 2, GPC3-O; 3, GPC3-O with HS oligosaccharide mixture; 4, cell surface GPC3-O conjugated with HS oligosaccharide mixture, and released by phospholipase C; 5, sample of line 3 purified by DEAE Sepharose chromatography; 6, GPC3-O conjugated with HS DP14; 7, cell surface GPC3-O conjugated with HS DP14 and released by phospholipase C; 8, sample of line 6 purified by DEAE Sepharose chromatography. This experiment was repeated three times with similar results. **c** Immunofluorescence detection of the GPC3 core protein and the attached HS chains. Cells were transfected with GPC3-O (3,4) or GPC3 control vector (1,2), and an hFGE vector. Two days after transfection, GPC3-O-expressing cells were incubated with (2,4) or without (1,3) bioHaHS. GPC3-O core protein and HS chains were detected with a mouse anti-GPC3 monoclonal antibody and TRITC-conjugated streptavidin, respectively. Scale bars: 40 μm. **d** Fluorescence quantification of biotin stained cells corresponding to **c** (1,2,3,4) by flow cytometry. Error bars represent means of triplicates ± SD. Source data are provided in a Source Data file.

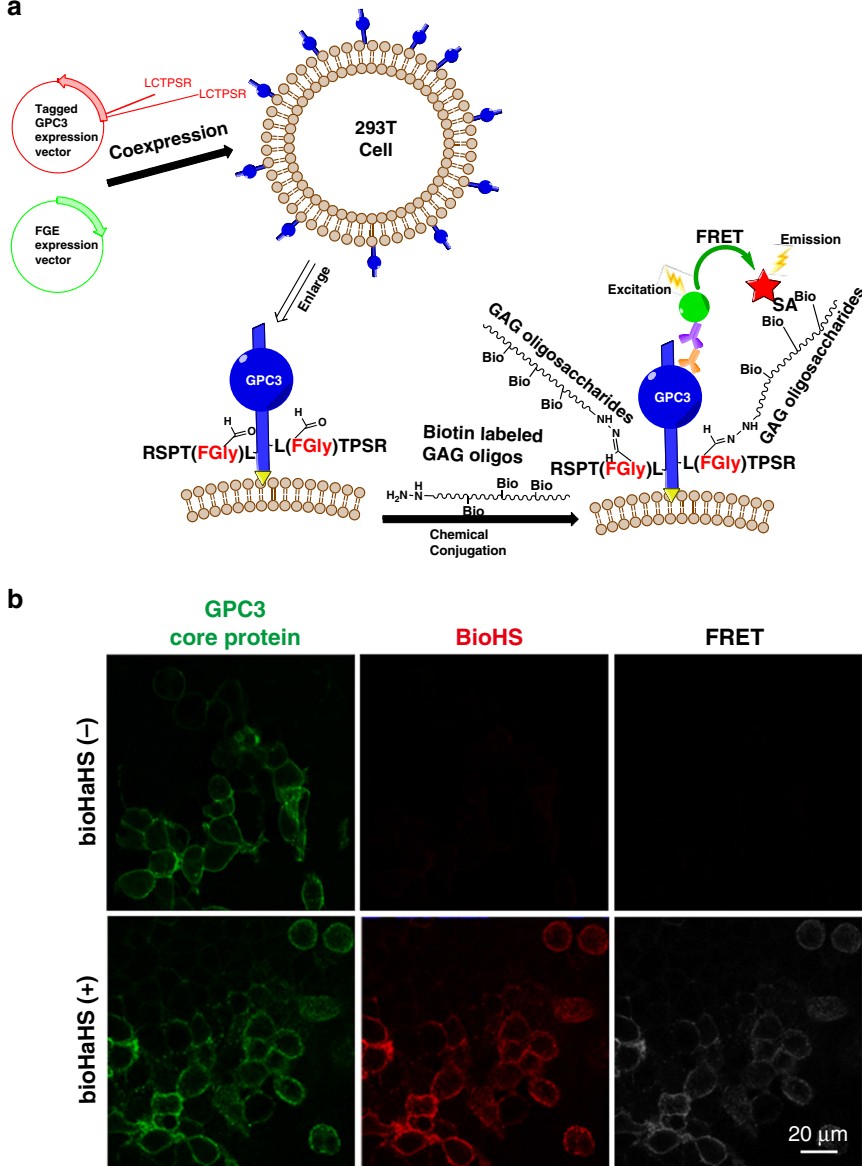

**Fig. 3 Intramolecular FRET between GPC3-O and attached HS. a** Diagram of the intramolecular FRET strategy for imaging HS side chains attached to GPC3-O. **b** FRET analysis of the GPC3 core protein and the attached HS chains. GPC3-O-transfected cells were incubated in the presence or absence of bioHaHS, and FRET was performed. This experiment was repeated three times with similar results. Scale bars: 20 μm.

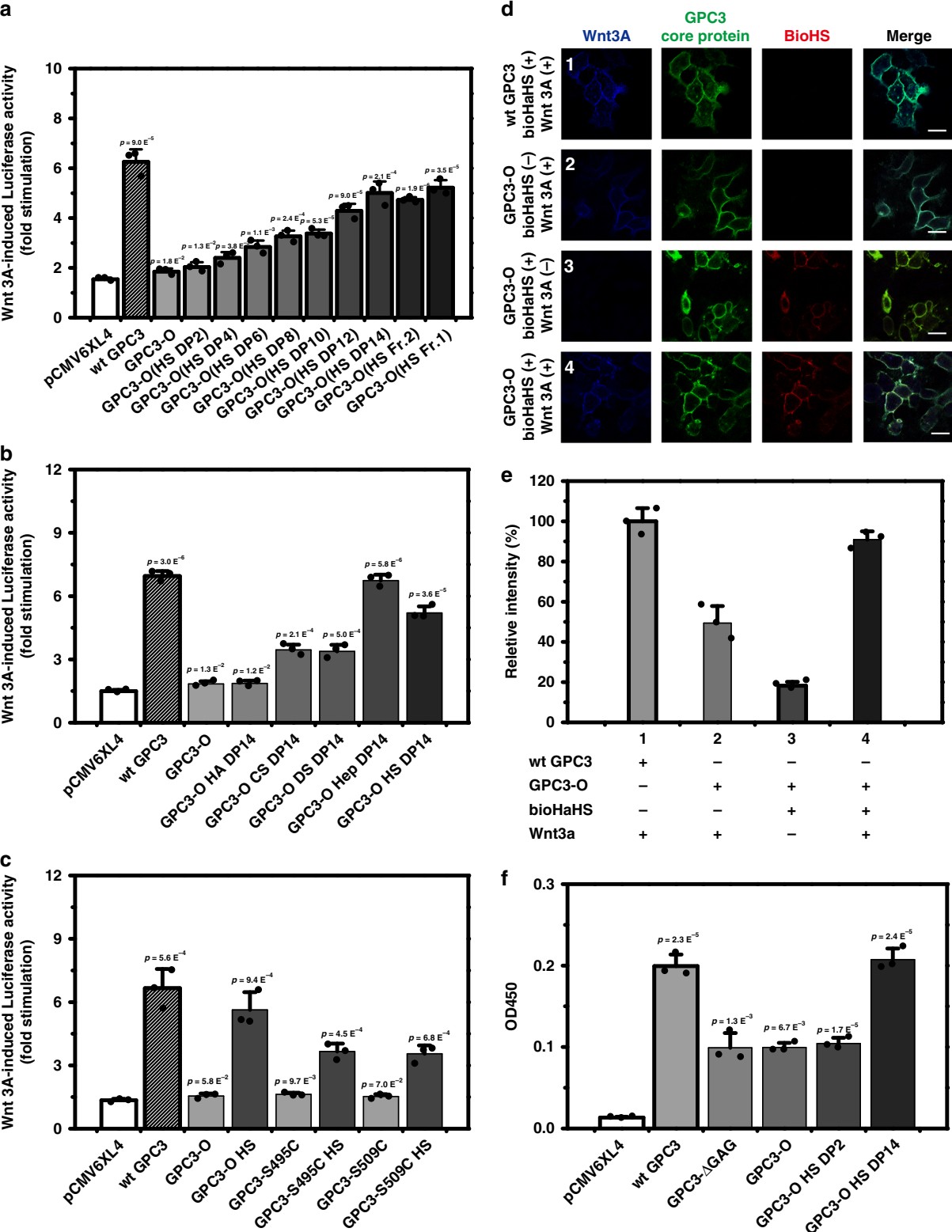

As shown in Fig. 4a, the addition of HS oligosaccharide chains to the GPC3 core protein increased the Wnt-stimulating activity and the degree of stimulation was proportional to the length of the attached HS oligosaccharides. In contrast, HA chains with various lengths did not promote Wnt3a activity (Fig. 4b and Supplementary Fig. 11a). CS and DS chains showed some stimulating activity when their length was DP8 or longer, but they were significantly less effective than the HS chains (Fig. 4b,

Supplementary Fig. 11b, c). Notably, the attachment of Hep chains stimulated Wnt activity more than the HS chains and the stimulation obtained with DP12 was as high as that of wild-type GPC3 (Fig. 4b and Supplementary Fig. 11d). These results indicate that the length and structure of side chains strongly affect the biological function of GPC3. On the other hand, the treatment of GPC3-O-transfected 293T cells with various GAG oligosaccharides without hydrazide labels did not show significant activation

**Fig. 4 The engineered GPC3 stimulates Wnt3a signaling.** Cells were transfected with the indicated GPC3 vectors, hFGE, and luciferase reporter, and β-galactosidase vectors. Two days after transfection, cells were incubated with the indicated HaHS. Cells were then incubated with control or Wnt3a CM and the luciferase and β-galactosidase activity were measured. Bars represent the fold stimulation induced by Wnt3a. **a**, Effect of the length of the HS oligosaccharides on Wnt3a cell signaling, DP2: disaccharide; DP4: tetrasaccharide; DP6: Hexasaccharide; DP8: octasaccharide; DP10: decasaccharide; DP12: dodecsaccharide; DP14: tetradecsaccharide; Fr. 2: oligosaccharide fraction longer than DP14 and shorter than Fr.1 fraction; Fr. 1: oligosaccharide fraction longer than Fr.2 fractio.n **b** Effect of the GAG type on Wnt3a signaling. **c** Effect of the number of GPC3 HS chains on Wnt3a signaling. **d** Binding of Wnt3a (1,3,4) (blue) to GPC3-(1) or GPC3-O (2,3,4) with (1,2,4) or without (3) bioHaHS oligosaccharides (red) expressing cells (green). GPC3-O-bioHaHS-transfected cells were treated with Wnt3a CM, and the cells were immunostained with rabbit anti-Wnt3a mAb and Alexa Flour 405-conjugated rat anti-rabbit antibody, mouse anti-GPC3 monoclonal antibody and FITC-conjugated goat anti-mouse antibody, and TRITC-conjugated streptavidin. 1, Wild-type GPC3 with Wnt3a CM; 2, GPC3-O with Wnt3a CM; 3, GPC3-O-bioHS with L CM; 4, GPC3-O-bioHS with Wnt3a CM. **e** Relative fluorescent intensity of Wnt3a bound to the surface of 239T cells treated with different conditions as indicated in **d**. **f** Binding assay between Wnt3a and the indicated GPC3 variants expressed in 293T cells. wt GPC3, wild-type GPC3. Error bars represent means of triplicates ± SD. *p*-values in all cases are compared with the corresponding negative control pCMV6XL4. Source data are provided in a Source Data file.

of Wnt3a signaling compared with activation by their hydrazide-labeled counterparts (Supplementary Fig. 12a), indicating that the attachment of GAG chains to core proteins is also critical for the Wnt-promoting activity of GPC3.

Next, we investigated whether the number of GAG oligosaccharides affects the activity of GPC3. To this end, we compared the Wnt3a-stimulating activity of GPC3 wild-type with that of the GPC3 mutant (GPC3-S495C or GPC3-S509C) with one side chain and two side chains. In all cases, we used DP14 oligosaccharides. The results showed that the activity of GPC3-O with two HS chains was similar to that of the GPC3 wild-type, and significantly higher than that of GPC3-S495C or GPC3-S509C with one HS chain (Fig. 4c), indicating that the number of HS side chains is an important factor for the function of GPC3, which is consistent with previous findings[26]. In addition, the activities of GPC3-S495C and GPC3-S509C were similar. In summary, Wnt signaling studies indicate that the engineered GPC3s display similar functions than the wild type.

We have previously shown that GPC3 binds to Wnt3a, and that the GAG chains are required for optimal binding[24,27]. We decided therefore to investigate whether the GAG chains can also increase the binding of the engineered GPC3 in the context of intact cells. GPC3-O-transfected 293T cells treated with or without bioHaHS oligosaccharides were incubated with Wnt3a-containing conditioned medium. After washing away the unbound material, we assessed the binding of Wnt3a to the cells by immunostaining with an anti-Wnt3a antibody. As shown in Fig. 4d, Wnt3a binds to the core protein of GPC3-O without the GAG oligosaccharide chains. This binding is significantly increased after the addition of HS oligosaccharide chains (Fig. 4d, e). The length of the HS chain is important for the binding of Wnt3a to GPC3, as no increased binding was observed with DP2, but DP14 produced a significant stimulation (Fig. 4f).

Next, we investigated the capacity of the engineered GPC3 to regulate the Shh signaling pathway. As the assessment of Hh signaling is generally performed in NIH3T3 mouse fibroblasts, we sought first to verify whether the engineered GPC3 can also be assembled on the cell surface of these cells. Immunofluorescent colocalization and FRET between the FITC-stained core protein and the TRITC-stained HS side chains were clearly detected on the surface of GPC3-O-transfected NIH3T3 cells after incubation with bioHaHS oligosaccharides (Fig. 5a, b). Next, the effect of specific GAG chains on the Shh-regulatory activity of GPC3 was studied. As shown in Supplementary Fig. 13a, the pretreatment of GPC3-O-transfected NIH3T3 cells with HaHS oligosaccharides dose-dependently strengthened the inhibitory activity of GPC3-O core protein on Shh signaling. Maximum inhibition was observed when the HaHS reached a 1 μM concentration. We also studied the effect of HS chain length on the GPC3-induced inhibitory activity on Shh signaling We found that this inhibitory activity gradually increased

with the length of HS side chains. The maximum inhibition was obtained with HS DP14 (Supplementary Fig. 13b), which is similar to what it was found for Wnt signaling. Interestingly, we also observed that the number of side chains had an impact on Shh signaling, since GPC3-O displaying two HS side chains had a stronger inhibitory activity than that of GPC3 with one HS chain (Supplementary Fig. 13c). Next, we studied the effect of different types of GAG side chains on the inhibitory activity of GPC3. The strongest effect was generated by HS and Hep, but CS and DS also resulted in significant inhibitory activity. These results indicate that the structure of the GAG side chains plays a key role in the inhibition of Shh signaling by GPC3 (Fig. 5c). Moreover, immunostaining also showed that Shh could bind to cells expressing GPC3 with or without HS side chains (Fig. 5d, e), which was consistent with results of previous studies[25]. Cells expressing GPC3 with HS chains showed more binding than those expressing the core protein. Overall, these experiments demonstrate that the engineered GPC3 mimics the inhibitory function of wild-type GPC3 on Shh signaling (Supplementary Fig. 12b).

**Use of the engineered GPC3 for the detection of GPC3-interacting proteins.** As described above, many of the interactions of PGs are mediated by GAG chains. However, some of these interactions could be difficult to characterize due to their low affinity. We have hypothesized that the ability to engineer PGs combined with the ability to metabolically label glycosylated proteins with monosaccharide analogs could provide a novel protocol to verify the potential interactions of PGs with glycosylated proteins. Metabolic labeling of glycans with unnatural monosaccharide analogs containing a reactive azido group has been widely applied for imaging and functional studies of glycosylated proteins. The azido group introduced into glycans can easily react with the alkyne group in various detection probes via click chemistry under physiological conditions[28].

We have previously used coimmunoprecipitation and cell-binding assays to demonstrate that GPC3 can interact with the Wnt receptors frizzleds (FZDs), such as FZD-4, FZD-7, and FZD-8[29,30]. This interaction is mediated by the GAG chains[29]. FZDs are cell surface glycoproteins. FZD-7, e.g., displays two N-linked glycosylation sites at N[63] and N[164]. Regardless of the final structure of the sugar chain, the conserved pentasaccharide core structure in N-glycans consists of two N-acetylglucosamines (GlcNAc) and three mannose residues[31]. Consequently, the azido analog of GlcNAc, N-azidoglucosamine (GlcNAz), can be introduced into the glycans of FZD-7 by metabolic labeling. On the other hand, alkyne-labeled GAG chains can be assembled into the GPC3 core protein using the method developed in this study. Thus, if the azido-tagged FZD-7 and the alkyne-tagged GPC3 are close enough on the cell surface due to their interaction, they could become covalently

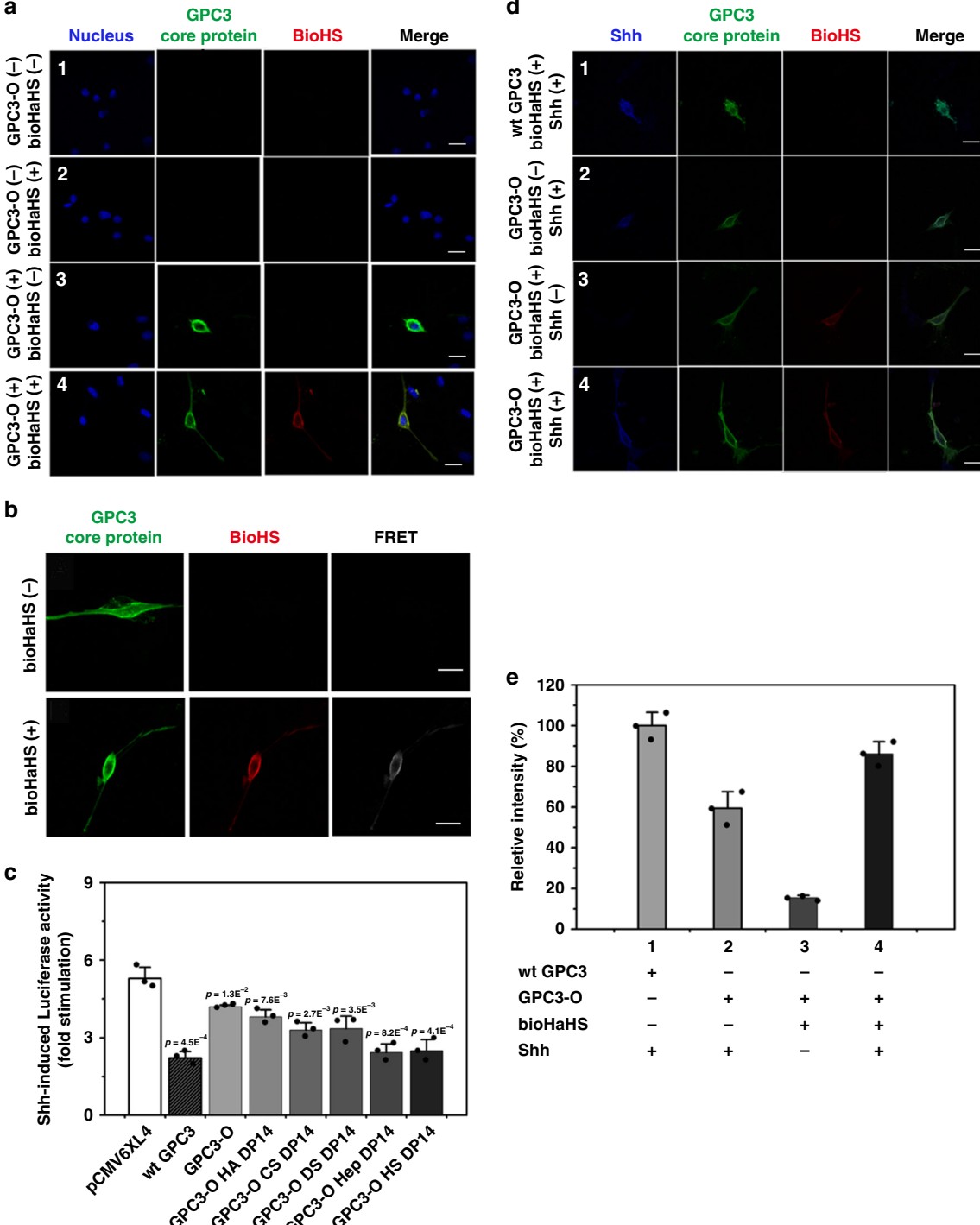

**Fig. 5 Engineered GPC3 inhibits Shh signaling. a** Fluorescence detection of GPC3 core protein and HS oligosaccharide chains in NIH3T3 cells. Cells transfected with engineered GPC3 (3,4) or negative controls (pCMV6XL4) (1,2) were incubated in the presence (2,4) or absence (1,3) of bioHaHS and were stained with a GPC3 antibody (GPC3) or TRITC-conjugated streptavidin. **b** Cells expressing engineered GPC3 were incubated with or without biotinylated HS, and FRET was performed. **c** NIH3T3 cells were transfected with GPC3-O or vector control (pCMV6XL4), and incubated with different types of GAG chains. A Hh luciferase reporter assay was then performed. Bars represent the fold stimulation induced by Shh. **d** Shh binds to GPC3-O-bioHS in the context of intact cells. 1, Wild-type GPC3 with Shh CM; 2, GPC3-O with Shh CM; 3, GPC3-O-bioHS with control CM; 4, GPC3-O-bioHS with Shh CM. **e** Relative fluorescence intensity of Shh bound to the surface of NIH3T3 cells expressing wild-type (wt) or engineered GPC3. wt GPC3, wild-type GPC3. All experiments were repeated three times with similar results. Error bars represent means of triplicates ± SD. *p*-values in all cases are compared with the corresponding negative control pCMV6XL4. Scale bars: 20 μm. Source data are provided in a Source Data file.

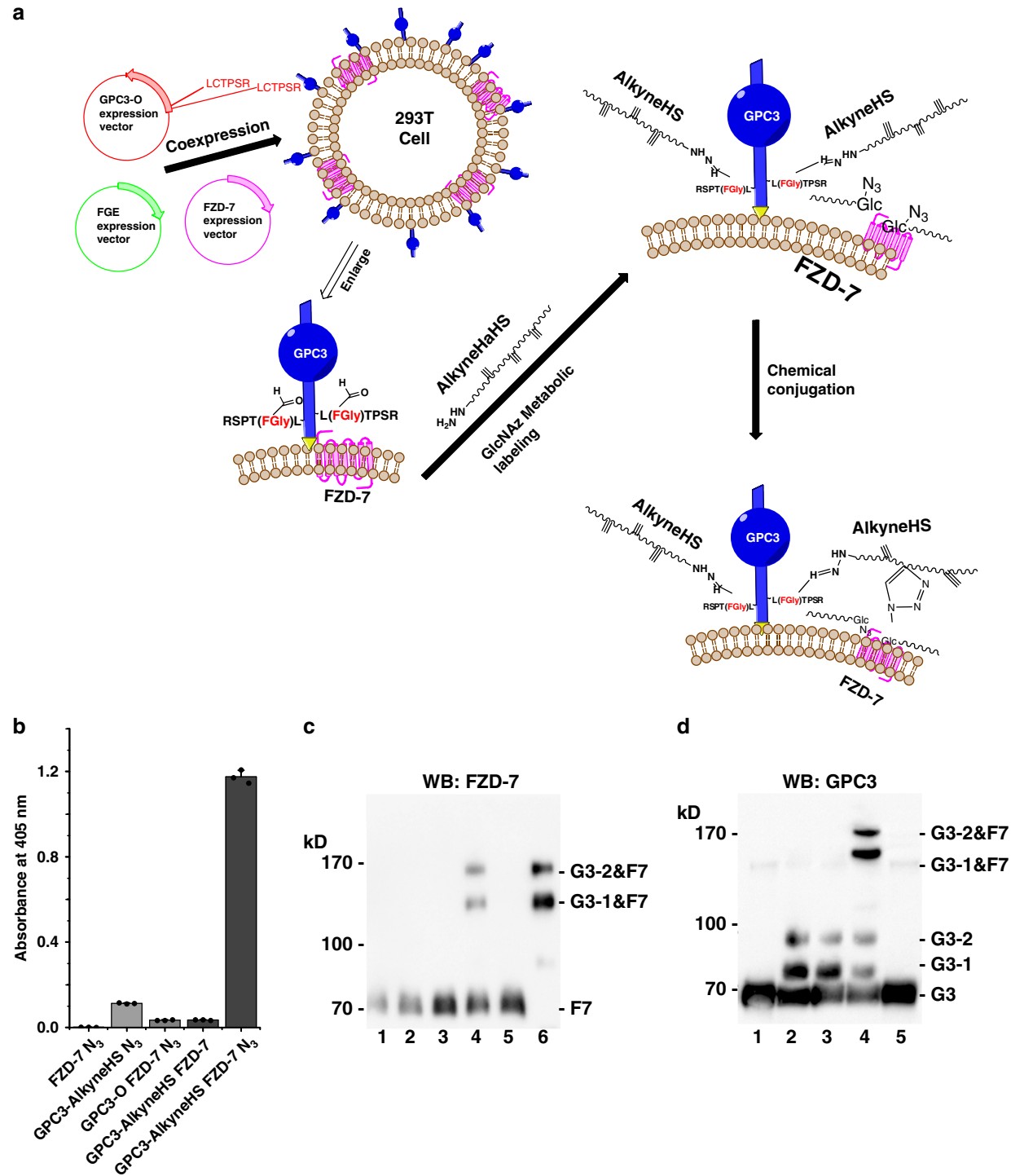

linked through click chemistry (Fig. 6a). To test this possibility, HEK 293T cells were co-transfected with GPC3-O, hFGE, and FZD-7 expression vectors, and the glycans were then metabolically labeled. Next, GPC3 was engineered by incubating the cells with alkyne-tagged HaHS oligosaccharides (alkyne HaHS) (Supplementary Fig. 3), which were prepared and verified as described in Supplementary Note 3. After the click chemistry reaction, cells were lysed with RIPA buffer and cell lysates were analyzed by an enzyme-linked immunosorbent assay (ELISA) and WB. For the ELISA, cell lysates were added to plates coated with an anti-GPC3 antibody. After the unbound material was

removed, an anti-FZD-7 antibody was added to detect FZD-7 bound to GPC3. As shown in Fig. 6b, a much stronger signal was detected for the binding of azide-labeled FZD-7 to GPC3-alkyneHaHS than for binding of unlabeled FZD-7 or negative controls. Notably, a low binding signal was observed when unlabeled FZD-7 was added to GPC3-alkyneHaHS. This could be due to the fact that GPC3/FZD-7 interaction is weak and can be disrupted by the RIPA buffer. These results suggest that the generation of engineered PGs with the protocol developed in this study could be useful to study protein–protein interactions mediated by GAG chains.

**Fig. 6 Engineered GPC3 interacts with FZD-7.** Cells transfected with GPC3-O and FZD-7 were incubated with alkyneHaHS and labeled with an azide group by metabolic labeling. The engineered GPC3 and FZD-7 were linked by click reaction. For the binding assay, cell lysates were added to wells covered with a GPC3 antibody, and the bound FZD-7 was detected using an anti-FZD-7 antibody. For the WB assay, protein samples were run through 8% SDS-PAGE and the transferred membranes were incubated with GPC3 or FZD-7 antibody and the corresponding secondary antibody. **a** Schematic of the experiment performed to detect the interaction between GPC3 and FZD-7. **b** Binding assay between GPC3 and FZD-7. Bars represent the absorbance at 405 nm (mean ± SD of triplicates). **c** Protein interactions analyzed by WB using FZD-7 antibody. Lane 1, cells were transfected with FZD-7 and wild-type GPC3, and then metabolically labeled with GlcNAz; Lane 2, cells were transfected with FZD-7 and engineered GPC3-O, metabolically labeled with GlcNAz, and then alkyneHaHS DP14 as side chains were attached to the GPC3-O on the cell surface; Lane 3, cells were transfected with FZD-7 and engineered GPC3-O, metabolically labeled with GlcNAz, incubated with HaHS DP14 without alkyne group, and then the click chemical reaction was performed; Lane 4, cells were transfected and metabolically labeled as in lane 2 and the click chemical reaction was performed; Lane 5, cell lysates prepared as in lane 3 were treated with heparinases before the click reaction; Lane 6, GPC3 and conjugated proteins in cell lysates generated as in lane 3 were purified with an affinity column with immobilized anti-GPC3 antibody. **d** Protein interactions analyzed by WB using GPC3 antibody. Lane 1, cells were transfected with FZD-7 and wild-type GPC3, and then metabolically labeled with GlcNAz; Lane 2, cells were transfected with FZD-7 and engineered GPC3-O, metabolically labeled with GlcNAz, and then alkyneHaHS DP14 side chains were attached to the GPC3-O on the cell surface; Lane 3, cells were transfected and metabolically labeled as in lane 2, and a click chemical reaction was performed; Lane 4, cell lysates prepared as in lane 3 were treated with heparinases before the click reaction; Lane 5, GPC3 and its conjugated proteins in cell lysates generated as in lane 3 were purified with an affinity column with immobilized with anti-GPC3 antibody; G3, core protein of GPC3-O; G3-2, GPC3-O with two HS DP14 chains; G3-1, GPC3-O with one HS DP14 chain; F7, FZD-7; G3-2&F7, the complex of GPC3-O with two HS DP14 chains and FZD-7; G3-1&F7, the complex of GPC3-O with one HS DP14 chains and FZD-7. All experiments were repeated three times with similar results. Source data are provided in a Source Data file.

As an additional approach to confirm the formation of a complex between azide-labeled FZD-7 and GPC3-alkyneHaHS, WB assays were carried out by using antibodies against FZD-7- or GPC3. As shown in Fig. 6c, treatment of anti-FZD-7 antibody detected two high molecular mass bands in the lysate of cells expressing both azide-labeled FZD-7 and engineered GPC3-alkyneHaHS, but did not detect these two bands in controls transfected with only one of the proteins or incubated with HaHS DP14 without alkyne group. The molecular masses of these two bands suggest that they represent complexes of FZD-7 with GPC3-O carrying one or two alkyneHaHS (DP14) chains. The corresponding complexes could be disrupted by pretreating the cell lysate with heparinases (Fig. 6c, lane 5), indicating that HS chains are required for the GPC3/FZD-7 interaction. Moreover, the GPC3/FZD-7 complexes could be purified from the cell lysate by using affinity chromatography for GPC3 (Fig. 6c, lane 6). Similar results were obtained when we used GPC3 antibodies for the WB analysis of cells expressing alkyne-tagged GPC3 and azide-labeled FZD-7. As shown in Fig. 6d (lines 1,2) the GPC3 antibody detected two narrow bands with molecular masses corresponding to complexes generated by the binding of GPC3 core proteins with one or two HS DP14 chains to FZD-7 (Fig. 6d, line 4). Furthermore, the bands corresponding to the GPC3-FZD-7 complexes could be disrupted by treatment with heparinases (Fig. 6d, lane 5) confirming the key role of the GAG chains in the formation of the GPC3-FZD-7 complex. These results provide additional support for the hypothesis that the generation of engineered PGs with the protocol developed in this study could be useful to study protein–protein interactions mediated by GAG chains of PGs.

## Discussion

GAG side chains play an irreplaceable role in the biological functions of PGs. However, the high degree of variability of the GAG structures makes it very difficult to study the relationship between these structures and their function. The availability of a methodology to precisely control the composition of the GAG chains would be extremely useful for the structure–function analysis of PGs. However, currently there is no proper method available to precisely control the structure of the GAG chains that are incorporated into the core proteins of PGs. In this study we developed a methodology that allows the addition of specific GAG structures to core proteins. To do this we took advantage of the ability of hFGE to specifically recognize a unique short

protein sequence (LCTPSR) and to catalyze the conversion of the Cys residue of this sequence to a fGly residue displaying an aldehyde group. A hydrazide group attached to a specific GAG chain can then quickly react with the aldehyde to form a relatively stable covalent bond under physiological conditions. We have demonstrated that this method can be used to engineer the assembly of PGs with various specific GAG chains in vitro and in vivo by biochemical and genetic manipulations. Most importantly, our study showed that an engineered GPC3 could perform similar biological functions to wild-type GPC3 in the regulation of Wnt and Hedgehog signaling.

Compared to most chemical and enzymatic approaches, methods using an aldehyde tag are a more convenient and robust approach for site-specific protein conjugation. This tagging methodology has been used for many different purposes such as fluorescence labeling, biotinylation, glycantation, drug-conjugation, fusion, immobilization and assembly of various proteins[22,32–36]. The efficiency of the conjugation is mainly determined by three factors: the conversion efficiency of Cys to fGly, the properties of molecules to be bound to the aldehyde-tagged protein, and the conditions of the conjugation reaction. The conversion of Cys to fGly in vivo is a cotranslational process catalyzed by FGE. This enzyme resides in the endoplasmic reticulum, and its levels can be dramatically increased by transfection with exogenous FGE[22]. The addition of copper ions was shown to improve the conversion efficiency[37], but in our study, this treatment did not significantly improve the assembly efficiency of the engineered GPC3 (data not shown). In contrast to other molecules conjugated to aldehyde-tagged proteins in previous studies[36], GAGs are more complex and highly negatively charged. Presumably, this would suggest that GAGs could be very difficult to conjugate. However, our results clearly show that GAG chains labeled with hydrazide at the reducing ends can efficiently be bound to AhGPC3 core proteins in cell lysates and on the surface of living cells. It should be noted that the conjugation efficiency of GAG chains to core proteins is strongly affected by the reaction conditions. We have found that a slightly acidic environment (pH 4–6) is necessary for high conjugation efficiency, which is consistent with a previous finding[38]. To reduce the detrimental effect of acidic conditions on cells, we kept a pH of 6.0 in the culture medium used during the conjugation step. Based on the results from the WB analysis, we estimate that in the cell lysate at least 60% of the GPC3 core proteins with one or two aldehyde tags could be conjugated with GAGs (Fig. 1b, c). Notably, our results

show that the engineered GPC3 can be detected on the cell surface for at least 24 h under normal cell culture conditions. This should provide enough time for functional studies. Two novel aldehyde-specific conjugation approaches have been recently developed[39–41]. Whether these new approaches result in a more efficient and stable conjugation between core proteins and GAG chains remains to be investigated.

GPCs are known to regulate cell signaling by interacting with various growth factors or morphogens as well as their receptors[42]. GPC3, in particular, has been shown to regulate canonical (β-catenin dependent) and noncanonical (β-catenin independent) Wnt signaling[43]. Previous studies showed that the core protein of GPC3 directly interact with Wnts[24], and that the HS side chains play an essential role in stabilizing the Wnt/GPC3/FZD complex by interacting with both Wnt and FZD[27,43–45]. However, the structural features of the GAG chains involved in the Wnt/GPC3/ FZD complex remained to be investigated. In the present study, the structural requirements of the GAG chains for effective canonical Wnt stimulation were investigated by using engineered GPC3 with different GAG structures. We found that the HS tetradecasaccharide is the minimal size required for optimal activity, although HS oligosaccharides from hexa- to decasacchacrides in size showed significant stimulation of signaling, which increased with increasing chain length. In addition, we established that GPC3 displaying HA oligosaccharide chains had no signaling activity, and that CS/DS oligosaccharide chains confer only a small stimulatory effect. Interestingly, Hep oligosaccharide chains show even stronger activity than HS oligosaccharides.

We have also demonstrated here that, similar to wild-type GPC3, the engineered GPC3 is able to inhibit Hh signaling[25]. The minimal size of the GAG oligosaccharide with the optimal inhibitory activity is a tetradecasaccharide, and both HS and Hep show the highest activity. Taken together, these results indicate that the structural features of GAG side chains strongly affect the function of GPC3.

In this study, we have also established a novel method for investigating the interaction of PGs with other proteins on the cell surface by combining the technologies of PG engineering assembly and metabolic glycan labeling. By using this novel method, we obtained a direct evidence of GPC3 interacting with FZDs, and confirmed that this interaction is mediated by HS chains.

In conclusion, this study establishes a versatile platform for the engineered assembly of PGs with specific GAG chains. Through further improvement and combination with other technologies such as metabolic labeling, this methodology will be very useful in the future for the study of the relationship between the structure and function of PGs in vitro and in vivo.

## Methods

**Materials**. SDS, PrimeSTAR™ HS, DNA polymerases, restriction endonuclease, and other genetic engineering enzymes were purchased from Takara, Inc. (Dalian, China). Chondroitinase ABC (EC 4.2.2.4), Heparinase I (EC 4.2.2.7) II and III (EC 4.2.2.8), HA from *Streptococcus equi* (15–30 kDa), Hep from porcine intestinal mucosa, HS from bovine kidney, phosphatidylinositol-specific phospholipase C from *Bacillus cereus*, and avidin from egg whites were obtained from Sigma. CS-A from whale cartilage and DS from porcine skin were obtained from Seikagaku Corp. (Tokyo, Japan). EZ-Link™ Biotin-LC-Hydrazide and *N*-azidoacetylglucosamine tetraacylated (GlcNAz) were obtained from Thermo Fisher Scientific. A novel GAG endo-lyase (HCLase) from a marine bacterium *Vibrio* sp. FC509 was prepared in our laboratory[46]. The anti-GPC3 mouse monoclonal antibody (αGCN) was also prepared in our laboratory[47]. The horseradish peroxidase (HRP)-conjugated goat anti-mouse (IgG) secondary antibody was purchased from Proteintech Group (Rosemont, USA). The anti-Wnt3a rabbit antibody, the anti-Shh rabbit antibody, the anti-FZD-7 rabbit antibody, the Alexa Flour 405-conjugated rat anti-rabbit (IgG) secondary antibody, the FITC-conjugated goat anti-mouse (IgG) secondary antibody, FITC-conjugated streptavidin, Cyanine3 azide, and TRITC-

conjugated streptavidin were obtained from Abcam (Shanghai, China). All other chemicals and reagents were the highest quality available.

**Cell lines and plasmids**. NIH3T3 and 293T cells were cultured in Dulbecco's modified Eagle's medium (DMEM) supplemented with 10% fetal bovine serum (FBS). L cells permanently transfected with Wnt3a-pLNCx or an empty vector (pLNCx) as a control were obtained from American Type Culture Collection and cultured in DMEM containing 10% FBS. To collect control or Wnt3a CM, the cells were grown at a high density for 4 days. All cell lines were maintained at 37 °C in a humidified atmosphere with 5% CO$_2$. Shh CM was generated by transfecting the Shh expression vector into 293T cells. The medium, which contained 2% serum, was collected on the sixth day after transfection.

Expression vectors for GPC3, GPC3ΔGAG, and Shh were described previously[25,28]. The amino acid sequences of GPC3 from F494 to C499 and G508 to G513 were singly or doubly mutated to the sequence LCTPSR specifically recognized by hFGE to convert Cys to fGly, three mutants were generated: GPC3-S495C and GPC3-S509C with a single aldehyde tag at Cys495 or Cys509, respectively, and GPC3-O bearing two aldehyde tags at both sites. These three GPC3 mutants were individually inserted into the expression vector pCMV6XL4. The full-length hFGE gene was synthetized and inserted into the expression vector pcDNA3.1a(+) by Genewiz, Inc. (Suzhou, China).

**Western blottings**. Transfected 293T cells were lysed for 30 min on ice by using lysis buffer containing 100 μM Mes, 0.1% SDS, 0.01% Triton X-100, and protease inhibitors (2 mM phenylmethylsulphonyl fluoride, 10 μg/ml leupeptin, and 10 μg/ml aprotinin) (pH 5.5). The hydrazide-labeled GAG (HaGAG) oligosaccharides were then added to the cell lysate and incubated at room temperature for 4 h. The protein samples were separated by SDS-polyacrylamide gel electrophoresis (PAGE) on an 8% gel and were then transferred to a polyvinylidene difluoride (PVDF) membrane (Millipore, USA). The membranes were blocked in blocking buffer (50 mM Tris-HCl, 150 mM NaCl, 0.1% Tween 20, and 5% skim milk, pH 8.0) for 1 h at room temperature and then incubated with 1 μg/ml αGCN overnight at 4 °C. After incubation with HRP-conjugated anti-mouse IgG secondary antibody for 1 h, protein bands were detected using enhanced chemiluminescence reagents (Proteintech Group, Inc.). The conjugation efficiency of the saccharide chains to the core proteins was determined by analyzing the relative intensity of the corresponding bands using ImageJ software (National Institutes of Health).

To confirm the binding of the HaGAG oligosaccharides to the GPC3 mutants expressed on the cell surface, 293T cells were transfected with a GPC3 mutant and an hFGE expression vector (2 : 1 wt/wt). Two days after transfection, HaGAG oligosaccharide chains (final concentration of 1 μM) were added to the medium (adjusted pH to 6.0) for 4 h. Then, the cells were lysed by lysis buffer and analyzed by WB as described above. To better show the conjugation of HaGAG oligosaccharides to GPC3 core proteins, we used DEAE Sepharose to concentrate the negatively charged PGs with HS chains in the cell lysate. Briefly, the DEAE Sepharose beads were added to the cell lysate and incubated at 4 °C for 4 h, the beads washed with 0.2 M NaCl in 50 mM NaH$_2$PO$_4$–Na$_2$HPO$_4$ (pH 6.0), and the bound material was then eluted with 2 M NaCl in 50 mM NaH$_2$PO$_4$–Na$_2$HPO$_4$ (pH 6.0). To extract cell surface GPC3, cells were washed twice with phosphate-buffered saline (PBS), then incubated with phosphatidylinositol-specific phospholipase C for 1 h at 37 °C, and after centrifugation the supernatant containing cell surface GPC3 was collected and analyzed by WB as described above.

**Assessment of the incorporation of the HaGAG oligosaccharide chains in living cells**. Cells were seeded on poly-L-lysine-treated coverslips and were transfected with a GPC3-O vector or a control vector, and an hFGE vector. Two days after transfection, bioHaGAG oligosaccharide chains were added to DMEM medium (adjusted pH to 6.0) and incubated for 4 h. Then, the cells were fixed with 4% paraformaldehyde in PBS for 20 min and blocked with 5% nonfat dry milk in PBS (blocking buffer) for another hour. After blocking, GPC3 and bioHaGAG chains on the cell surface were detected by using αGCN with a FITC-conjugated anti-mouse IgG secondary antibody and TRITC-conjugated streptavidin, respectively. The cells were washed three times with PBS after each step. To assess the conjugation between bioHaHS and AhGPC3, HEK 293T cells were seeded on 24-well plates and co-transfected with GPC3-O or a control vector, and an hFGE vector. Two days after transfection, cells were incubated with or without bioHaGAG for 4 h in DMEM medium (adjusted pH to 6.0). The cells were suspended, collected, washed, blocked with with 5% nonfat dry milk in PBS, and incubated with FITC-conjugated streptavidin for 1 h on ice in the dark. Finally, the cells were washed twice with PBS and analyzed on a Backman FC500 Flow Cytometer. Each experiment was performed thrice in triplicates.

**Wnt3a reporter assay**. Cells were seeded on a 24-well plates and co-transfected with the indicated expression vectors, a luciferase reporter plasmid driven by the TOPFLASH promoter that contained several TCF/β-catenin-responsive elements[48], and a β-galactosidase expression vector. The transfection was performed by using Lipofectamine 3000 (Invitrogen). Twenty-four hours after transfection, HaGAG oligosaccharides or GAG oligosaccharides without hydrazide labels were

added to the medium (adjusted pH to 6.0) for 4 h and the cells were incubated for another 24 h with L or Wnt3a CM. Finally, cells were lysed and luciferase activity was measured according to the neolite Reporter Gene Assay System protocol (PerkinElmer). Transfection efficiency was normalized by measuring β-galactosidase activity. Each experiment was performed at least three times in triplicates.

**Binding assay between assembled GPC3 and Wnt3a.** HEK 293T cells were seeded on six-well plates and co-transfected with the GPC3-O vector or a control vector, and an hFGE vector (2:1 wt/wt). One day after transfection, HaHS oligosaccharides were added to new medium (adjusted pH to 6.0) at a final concentration of 1 μM. After incubation overnight, the cells were lysed using RIPA lysis buffer containing protease inhibitors (2 mM phenylmethylsulphonyl fluoride, 10 μg/ml leupeptin, 10 μg/ml aprotinin) for 30 min on ice. For the binding assay, a 96-well plate was coated with a GPC3 mAb (aGCN) (0.4 μg/well) at 4 °C overnight. After blocking with blocking buffer (5% skim milk in PBS), appropriately diluted cell lysates were added to each well and incubated at room temperature for 2 h, and then the plates were incubated with L or Wnt3a CM at room temperature for another 2 h. Then, each well was washed three times with PBST (phosphate buffered saline with Tween 20) to remove any nonspecific binding. Finally, bound Wnt3a was detected using an anti-Wnt3a rabbit antibody, followed by incubation with an HRP-conjugated goat anti-rabbit IgG and 3,3′, 5,5;-tetramethylbenzidine (TMB) as a substrate.

**Hh-reporter assay.** NIH3T3 cells were plated into six-well plates and co-transfected with an Hh-reporter vector, a β-galactosidase expression vector, and the indicated expression vectors using Lipofectamine 3000 (Invitrogen). Twenty-four hours after transfection, cells were transferred to 24-well plates at 50% confluence, and on the next day, HaGAG oligosaccharides or GAG oligosaccharides without hydrazine labels were added to the medium (adjusted pH to 6.0) to a final concentration of 1 μM and incubated at 37 °C for 4 h. Then, Shh CM or control CM was added and cells were incubated for another 48 h. Finally, cells were lysed and luciferase activity was measured and normalized by the corresponding β-galactosidase activity. Each experiment was performed at least three times in triplicates.

**Binding of growth factors to the cell surface.** Cells were seeded on poly-L-lysine-treated coverslips and transfected with the GPC3-O vector or a control vector, and an hFGE vector. Two days after transfection, bioHaGAG oligosaccharides were added to the medium (pH adjusted to 6.0) for 4 h. Then, the cells were incubated with ice-cold DMEM containing 20 mM HEPES pH 7.4, and 0.1% bovine serum albumin for 1 h at 4 °C. Cells were then treated with Wnt3a, or Shh or control CM for another hour at the same temperature. To assess cell surface binding, unbound ligand was removed by washing three times with ice-cold PBS and the cells were fixed with 4% paraformaldehyde in PBS for 20 min. For immunostaining, the cells were blocked with 5% nonfat dry milk in PBS (blocking buffer) for another hour. After blocking, Wnt3a, GPC3, Shh, and bioHaGAG oligosaccharides were detected with an anti-Wnt3a rabbit antibody and an Alexa Flour 405-conjugated rat anti-rabbit antibody, a αGCN antibody and a FITC-conjugated anti-mouse IgG secondary antibody, an anti-Shh rabbit antibody and an Alexa Flour 405-conjugated rat anti-rabbit antibody, and TRITC-conjugated streptavidin, respectively. The cells were washed three times with PBS after each incubation step to remove nonspecific interactions. Image analysis was performed using the ImageJ program (National Institutes of Health) as previously reported[49]. In brief, a mask, which was constructed by manually outlining cells in the image to measure the fluorescence intensity. The mean fluorescence intensity of other regions of the cell was obtained by measuring several representative regions by moving the mask. The fluorescence intensities corresponding to Wnt3a or Shh were calculated after subtracting the background fluorescence. The staining intensity of Wnt3a or Shh bound to the surface of indicated cells is shown as relative values against that of cells expressing wild-type GPC3.

**Interaction assay for GPC3 and FZD-7.** HEK 293T cells were seeded on 10 cm plates and co-transfected with the GPC3-O vector or a control vector, an hFGE vector and an FZD-7 vector (2 : 1 : 2 wt/wt/wt). One day after transfection, GlcNAz was added into the culture supernatant at a final concentration of 40 μM. Twenty-four hours later, alkyneHaHS DP14 chains and GlcNAz were added to new medium (pH adjusted to 6.0) at final concentrations of 1 μM and 40 μM, respectively. After incubation overnight, a click reaction was performed as follows: the cells were incubated with PBS containing 250 mM tris-(benzyltriazolylmethyl) amine, 50 mM CuSO₄, and 500 mM L-ascorbic acid sodium salt (SA) for 2 h at room temperature[50]. Then, the cells were lysed using RIPA lysis buffer containing protease inhibitors (2 mM phenylmethylsulphonyl fluoride, 10 μg/ml leupeptin, 10 μg/ml aprotinin) for 30 min on ice.

For the binding assay, a 96-well plate was coated with a GPC3 mAb (aGCN) (0.4 μg/well) at 4 °C overnight. After blocking with blocking buffer (5% skim milk in PBS), appropriate diluted cell lysates were added to each well and were incubated at room temperature for 2 h. Then, each well was washed three times with PBST containing 1 M NaCl to remove any nonspecific binding. Finally, bound FZD-7

was detected using an anti-FZD-7 rabbit antibody, which was followed by incubation with an HRP-conjugated rat anti-rabbit IgG antibody, and TMB as substrate.

For the WB assay, the protein samples were separated by SDS-PAGE on an 8% gel and transferred to a PVDF membrane. The membranes were blocked in blocking buffer for 1 h at room temperature and then incubated with αGCN or FZD-7 rabbit antibodies overnight at 4 °C. After incubation with the corresponding HRP secondary antibodies for 1 h, protein bands were detected using enhanced chemiluminescence reagents (Proteintech Group, Inc.).

**Statistics.** Statistical analyses were performed using Origin 8.0. Each experiment was done at least three times by triplicates. Error bars represent means of triplicates ± SD. For comparison of the statistical differences between two groups, Student's t-test (two-sided) was carried out after comparing the variances of a set of data by F-test.

**Reporting summary.** Further information on research design is available in the Nature Research Reporting Summary linked to this article.

## Data availability
All data supporting the findings of this study are available within the paper (and its Supplementary Information files). All relevant data generated during this study or analyzed in this published article (and its Supplementary Information files) are available from the corresponding author on reasonable request. Source data are provided with this paper.

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

## Acknowledgements

This work was supported by the National Natural Science Foundation of China (numbers 31570071, 31971201, and 31800665), the National Natural Science Foundation of Shandong Province (number ZR2018BC013), the Science and Technology Development Project of Shandong Province (number 2018GSF121002), the General Financial Grant from China Postdoctoral Science Foundation Grant (number 2017M622189), the Project of Taishan Industry Leading Talent of Shandong Province (tscy20160311), and the Major Scientific and Technological Innovation Project (MSTIP) of Shandong Province (2019JZZY010817). We thank Haiyan Yu, Xiaomin Zhao, Zhifeng Li, Jingyao Qu, and Jing Zhu from Life Science General Research Technology Platform of SKLMT (State Key Laboratory of Microbial Technology, Shandong University) for the assistance in laser-scanning confocal microscopy, flow cytometry, and NanoLC-MS/MS experiments.

## Author contributions

W.W. designed and performed the immunofluorescence assay, the Wnt3a and Shh reporter assay, and the analysis of the interaction between GPC3 and FZD-7. N.H. prepared the hydrazide-GAG oligosaccharide fractions. Y.X. assisted in establishing the expression vectors for GPC3 mutants. Y.Z. performed flow cytometry assays. L.S. designed and performed western blot assays. J.F. provided some important suggestions for the experimental design and contributed to the writing of the manuscript. F.L. conceived the project, designed the experiments, and wrote the manuscript.

## Competing interests

The authors declare no competing interests.
