## [Peer Review File · Nature Communications]

Reviewers' comments:

Reviewer #1, an expert in glycobiology and chemical biology (Remarks to the Author):

This is an interesting manuscript describing novel methodology for addition of GAG oligosaccharides and chains to proteoglycan core proteins, via engineering of sites within the core protein sequence for addition of an aldehyde-bearing formylglycine residue. This allows covalent attachment of oligosaccharides or chains reducing-end labelled with a hydrazide or other related groups which will readily react with the aldehyde under mild conditions compatible with living cells. This permits production of PG core protein labelled with specific GAG moieties, and subsequent evaluation of cell functions mediated by the GAG moieties attached to the core proteins. The authors exemplify the approach by focussing on glypican 3 (GPC3), and its effects on Wnt and Hedgehog signaling. The engineered constructs behaved in the expected manner based on previous data for these interactions. They also demonstrate an additional approach for assessing interactions of PGs with targets which are glycoproteins, through metabolic labelling of glycans with an azido function, that can be tested for coupling with alkyne-labelled GAG moieties on engineered PG core proteins. This was exemplified by studying azido-labelled Frizzled receptor interaction with alkyne-labelled GPC3 GAG chains.

Overall these are valuable new methodologies which make an important contribution, and open up a lot of new and interesting experimental possibilities for understanding the biology of PGs, and in particular the structure-activity relationships in a true cellular context. However, there are a number of issues with the manuscript in its current form which need to be addressed before I would recommend publication.

Major issues:

1. There are difficulties in understanding the experiments due to poor descriptions and varied nomenclature for different reagents (eg. HaGAG, bioHaGAG, bioHS, HaHS, HS-bio, alkyneHaHS etc). It would be a good idea to add a panel in a figure showing schematically what all these different constructs are, to help the reader follow the experimental rationale. In general the authors need to carefully check their use of terminology in the Results and figure legends, to improve clarity/accuracy.

2. Results, p5 and Fig 1: the decreasing efficiency of conjugation of oligosaccharides, and the difficulty in getting 2 oligosaccharides attached, is a concern. The authors should provide some information on efficiency of intact chain attachment – is it even lower than the figures for DP15 saccharides? In general this raises a question about whether the statement that “oligosaccharides can be efficiently linked to the aldehyde-tagged core protein” is accurate.

3. Results, p6: the authors contend that the amount of unmodified core protein is higher in the experiments in living cells, and may be due to cytoplasmic PG not accessible to the labelled oligosaccharides. This question should be checked by specifically analysing cell surface PGs stripped from the cell surface for example using proteases.

4. Results p6: the authors describe using biotinylated HaGAG oligosaccharides (bioHaGAG), but there is no clear description of how these are made. The Methods describes an approach for double labelling GAG chains with hydrazide and biotin, but this is inadequately described and confusing, and it's not clear how dual labelled constructs are obtained (reducing end and also intra-chain label ??) and raises questions about the data in Fig S5 and the usage of these constructs. The validation of these dual-labelled constructs needs to be added.

5. Results, p13: experiments with azido-tagged Frizzled – can the authors rule out the possibility that the azido-tagged Frizzled reacts directly with aldehyde-tagged GPC3?

Fig 5, panel A, the schematic needs to better describe that dual-tagged GAG (with hydrazide and

alkyne tags) is needed, and also show the Frizzled as being azido-tagged.

6. The approaches described require complex multiple co-transfections of cells (as many as 4 constructs), but no information is provided about the % transfection efficiencies, or how differential transfection will affect the outcome of experimental usage of the resulting mixed cell cultures. Indeed, the immunofluorescence data seem to show such variation.

7. Some of the background experiments to validate hydrazide labelling are insufficient and not well described. From Fig S4 it seems that the only measure for % labelling for larger saccharides is claimed resistance of the reducing terminal tetrasaccharide to cleavage by heparin lyases. More information by other methods is needed to validate the labelling.

Fig S4 is also missing information on panel C and the MS data is not convincing (shows just a series of spikes at high m/z and no information on background noise).

8. It would be useful in the Discussion for the authors to discuss more about the limitations which the techniques are subject to (eg. chain size, numbers of chains for PG cores with multiple chains etc), and potential further improvements to increase efficiency etc, and wider applications of the techniques.

Minor issues:

1. Abstract: the authors should be careful to indicate that both GAG oligosaccharides and GAG chains can be attached using the methodology. In fact most (but not all) of the data is achieved with oligosaccharides. In general in the text the authors need to be more clear in describing oligosaccharides vs chains in the various experiments.

2. Results, p6: reference to Fig S4 should be S5 ?

3. Results, p8: the details of how DEAE chromatography was used needs to be added. Use of the term HS-Bio needs to be better defined. For Fig 3a the text says GAG chains, but in fact oligosaccharides are used.

4. Fig 3, legend: panel A need to define better which oligosaccharides and "chains" are used. Fr 1 and 2 need to be properly defined and their source and preparation given in the Methods.

5. Fig 4, legend: panel A, the use of HS-Bio is confusing since the authors seem to be using TRITC-streptavidin to detect biotinylated HS.

6. Fig S7, legend: what concentrations of saccharides are used in panels b and c ? Fr1 and Fr2 saccharides are not defined.

7. There are large numbers of typos and poor grammar usage which need to be corrected to improve the clarity of the writing.

Reviewer #2, an expert in proteoglycan engineering (Remarks to the Author):

The manuscript by Wang and coworkers describes a method for generating semi-synthetic proteoglycan structures directly at the surface of live cells. Proteoglycans are key regulators of growth factor signaling activity in cells linked to their proliferation and differentiation. They are of great interest as targets to manipulate cellular activity but have been very difficult to study due to the complexity and non-templated biosynthesis of their glycosaminoglycan (GAG) modifications. Many efforts have been developed to gain control over the GAG component of cellular signaling,

some are cited in this paper; however, the ability to control GAG glycosylation patterns of individual proteoglycans on the cell surface has not been achieved. In this context, the present work constitutes a significant step forward in this area. The authors approach the problem of controlling PG glycosylation by taking advantage of the aldehyde tag technology developed by Bertozzi. By replacing the native glycosylation sites in the proteoglycan, glypican 3 (GPC3), with short sequences containing cysteine, which can be converted into formyl glycine residues by the action of the formyl glycine modifying enzyme, the authors introduced new sites for chemical conjugation of hydrazide modified HS chains with well-defined structures directly at the cell surface. They then proceed to evaluate the activity of their neoglycoconjugates in signaling through Wnt and SHh proteins and use a combination of metabolic oligosaccharide engineering and click chemistries to capture the signaling complexes formed in the presence of the engineered PG structures. This study is very promising with potentially high impact on the field; however, there are some significant experimental issues and problems with data interpretation, due to lack of appropriate controls, that raise questions about the conclusions of this study. While my concerns regarding the described experiments and conclusions are described in detail below, the most critical issues the authors need to address prior to publication, in my opinion, are the harsh conditions (prolonged exposures of cells to low pH environments) needed to construct the neoglycoconjugates on the surfaces of living cells in the absence of assessment of cell viability and effects on cell function, the inefficiency (or lack of quantification) of the conjugation process, and the lack of important controls in the signaling and crosslinking assays that bring the validity of the conclusions into question. Following are my specific comments regarding the different aspects of this manuscript:

1. Generation of hydrazide-labeled GAGs. The manuscript text or the methods section (page 23) does not specify the pH of the buffer used for HS labeling with adipic hydrazide. The efficiency of GAG functionalization was determined either by purification of the conjugates for short disaccharides or by Hepase III digest in combination with MS analysis for longer oligosaccharides. The compositions of the oligosaccharides after digest were confirmed by disaccharide analysis. I wasn't able to find the SEC traces for the purified disaccharide conjugates, only the crude mixtures before purification were presented. The traces for the purified GAGs should be included in the SI. It is a bit challenging to follow all the different GAGs generated and used in the study. The authors should include a table in the SI summarizing all of the structures, their disaccharide composition, and the yields for hydrazide conjugation.

2. Generation of GPC3 mutants. The text on page 4 describing the generation of mutants and data in Fig 1 and S1 do not specify what cells these mutants were expressed in. Is it HEK293T cells as described later in the text?

I assume HEK293T cells don't express endogenous GPC3, can the authors confirm that? This should be clarified earlier in text and in Figures 1 and S1.

Expanded western blots in Fig 1b/c (full length) for all mutants and wild type GPC3 should be included in SI and it should be clarified what GPC3 is. Is this wild type fully glycosylated GPC3 expressed by the HEK293T cells?

What was the efficiency of conversion of Cys to fGly? Some data are presented in Fig S1 but it is unclear how efficient the process is overall and particularly for GPC3-O. MS data should be included.

The use of the term glycanation, which is not the same as glycosylation, in the text and figures should be checked and corrected.

3. In vitro GAG conjugation to GPC3 mutants. The authors state that the double mutant GPC3-O was conjugated with one and two chains in a ratio of 50% to 20% due to close proximity of the

two aldehyde tags, with 30% of the protein remaining unconjugated. It is possible, but this can also be caused by incomplete Cys-to-fGly conversion in the aldehyde tag. In general, the conjugation of hydrazide-labeled HS to the mutants seems to proceed, but the question is what the overall efficiency of the process is. Quantitative analysis of the Cys-to-fGly conversion by MS would confirm the authors' conclusion regarding steric effects or would give clear indication as to the efficiency of the process.

In Figure 1b/c, does the first GPC3 band correspond to fully glycosylated wild type GPC3 expressed in the HEK293T cells? Or is it one of the mutants treated with an oligosaccharide mixture? This should be clarified and if the latter is the case, a fully glycosylated wt GPC3 control expressed in HEK293T cells should be included for comparison.

The authors also generated the Cys509 mutant, but HS conjugation data is not provided in Fig 1 or in SI. The Cys509 mutant was used in cell signaling assays in this study but data for hydrazide conjugation to this protein in vitro or on cell surfaces was not provided. Is conjugation to this site as efficient as to fGly 459? Is the yield of conversion of Cys509 to fGly509 the same for Cys459 to fGly459? Could differences in fGly generation at these two sites explain why the doubly HS modified GPC3-O forms less efficiently?

4. Cell surface HS conjugation to GPC3 mutants. The hydrazide conjugation reaction requires low pH to proceed and, accordingly, the authors subjected the cells for prolonged periods of time 1-8 hrs to pH 6 conditions. An obvious and significant concern is the damage to the cells under those conditions. No data is provided regarding the cells' viability and proliferation capacity after the treatment. Since the authors evaluate sensitive signaling responses after such a treatment, these assays with appropriate controls should be included.

The authors should quantify the extent of HS conjugation to the GPC3 mutants at the cell surface. The western blot in Fig 2B shows that the process is clearly inefficient and concentration of the conjugated proteins by ion exchange was needed to for the conjugates to clearly manifest on the blot. Perhaps this can be addressed by using a different conjugation method not requiring low pH conditions.

Page 6, line 3 states that the HS GAGs were conjugated to the transfected cells under optimized conditions in SI Fig S4. This figure shows SEC analysis of hydrazide HS. The text should specify what the optimized conditions are (presumably, 1 μ M HS, pH 6 for 6 hrs at 37 deg C??).

Fig S5c describes the surface residence time of the conjugates (labeled as "vanishing time"). Is the loss of surface HS due to endocytosis by the cells or hydrolysis of the hydrazide linker? What conditions (ie, pH) is this experiment performed in? Assessment of the stability of the linker, which is prone to hydrolysis, should be provided.

Fig 2: the fluorescence micrographs in the figure are rather small and hard to evaluate visually. The authors should consider adding to the SI quantitative analysis for immunofluorescence and FRET efficiency as well as higher resolution images.

The use of biotinylated HS chains to monitor conjugation to the GPC3 mutants on the cell surface provides evidence for this process proceeding as described; however, it's difficult to tell from the fluorescence micrograph in Fig 2c what the background binding of hydrazide-HS is to the surface of control cells treated with the empty vector. Those controls were not included in the western blots in Fig 2b. A more sensitive and quantitative assessment of conjugation using flow cytometry might be beneficial here. Also, probing for sufficient HS conjugation more directly by using standard HS-binding proteins such as FGF2 or the anti-HS antibody 10E4 instead of relying on biotin-streptavidin interaction may be beneficial. Although endogenous HS on the cell surface may pose a challenge in such an experiment.

5. Signaling response in GAG-engineered cells. The observations of increasing Wnt activity in response to increasing HS length is interesting. The authors should comment on the influence of conjugation efficiency on signaling activity, since they indicated that the reaction yield decreases with increasing HS chain size. This may mean that high levels of glycosylation may decrease Wnt activity. In these experiments, soluble HS lacking the hydrazide unit should be included. The role of HS in Wnt signaling can be complex and is often poorly defined. For instance, cell surface HS can inhibit Wnt signaling by sequestering the protein away from its receptors, while soluble heparin has been shown to promote Wnt association with cognate receptors and activate signaling. It appears that the signaling assay was performed without washing the excess soluble hydrazide-HS. It is not possible to tell from these experiments if the changes in signaling through Wnt (or SHh) in Figure 3 (and 4) were the result of activity from cell-surface or the unwashed soluble HS chains.

The authors should also clarify whether GPC3 is the fully glycosylated wild type protein? In most of the figures it's referred to as "GPC3 core protein" but it is not clear if it is glycosylated. The presence of endogenous HS on the surface of the HEK291T cells and its possible effects on growth factor binding and activity should be discussed in the manuscript.

The harsh acidic treatment of cells for prolonged periods of time is of concern. The authors should clarify if the included controls were subjected to the same treatment and how that affected signaling responses with respect to cells treated under physiological conditions.

The authors indicate, based on fluorescence immunostaining data in Fig 3, that Wnt3a binds to GPC3 in the absence of HS modifications. Did the authors consider that Wnt3a might bind to endogenous HS proteoglycans on the cell surface besides GPC3, which would be expected in cells with fully functional HS biosynthesis? What is the evidence that Wnt3a binds to the non-glycosylated GPC3 and not another endogenous HSPG on the cell surface?

The SHh signaling experiments suffer from the same technical issue as the Wnt signaling experiments, which is the presence of soluble unreacted hydrazide-HS during the course of the assay. Soluble HS can inhibit SHh signaling and may be responsible for the observed inhibitory activity (including dose dependence). The authors need to demonstrate that it is not the soluble or otherwise adsorbed HS that is responsible for the observed changes in signaling (negative controls consisting of cells treated under identical conditions with HS lacking the hydrazide functionality should be included).

The final part of the paper describing the capture of GPC3 with alkyne-modified HS and FZD-7 with azide-modified N-glycans through proximity induced cycloaddition reaction in the absence of copper catalyst is not convincing. Proximity induced alkyne-azide cycloaddition, while demonstrated early on by Sharpless in catalytic sites of enzymes, is rarely observed in the case of relatively weak interactions, such as those between GAGs and GF/receptor complexes. The included controls using cells expressing only one of the protein components do not provide strong enough evidence that covalent crosslinking is indeed occurring in this system. Controls performed with metabolically labeled FZD-7 and GPC3 modified with HS without alkynes were missing in Fig 5. What is the rationale for the core or LacNAc extensions of the N-linked glycans to be in close proximity to the HS chain? What is the efficiency of GlcNAz labeling of N-glycans in FZD-7? In general, GlcNAz incorporation into N-glycans is not as effective as the incorporation of GalNAz, which undergoes efficient epimerization to produce UDP-GlcNAz.

Reviewer #3, an expert in proteoglycan signalling (Remarks to the Author):

In this study, using glypican-3 as a model, the authors applied an aldehyde tag-based approach to assemble PGs with GAG chains, including different types GAG and different sized HS. Functional studies showed that the engineered glypican-3 can regulate Wnt and Hedgehog signaling like the wild type, and the regulation mainly is mediated by assembled HS, not by other GAGs. More HS focused study further showed that the regulatory activity of engineered glypican-3 on Wnt and Hedgehog signaling depends on size and chain number of assembled HS. In combination with metabolic glycan labeling, this approach was extended to study and confirmed that glypican 3 interacts with Frizzled 7. In summary, this aldehyde tag-based approach has been clearly shown to be applicable for the structural and functional studies of proteoglycans with specific GAG chains.

Major concern:

1. Technically not novel: The aldehyde tag-based assembly of carbohydrate moiety to glycoprotein and metabolic glycan labeling both have been well established and widely used in glycobiology. This study applies this approach to study proteoglycan.
2. Application study did not result in any new discovery, only confirmed reported findings in literature.
3. Statistical analyses are not clearly stated: Which tests were used ?, how many repeats for each figure ? Except Fig.4C, no p values for all other bar figures are included.

Dear Reviewers:

Thank you for reviewing our manuscript (NCOMMS-20-03326) entitled “**Assembly of Proteoglycans with Specific Side Chains on Living Cell Surface and Application in Interaction with Their Target Proteins**”. We greatly thank the positive comments and the recommendations of publishing our manuscript in Nature Communications.

According to the suggestions and comments, we have performed some additional experiments and revised the manuscript. Specifically, (a) the tryptic fragments of GPC3-O were analyzed on a linear ion trap-Orbitrap hybrid mass spectrometer coupled with a nano-LC system to demonstrate “the efficiency of conversion of Cys to fGly” (Fig. S2), (b) the “hydrazide labelling for larger saccharides” was confirmed alternatively by analyzing hydrazide-labeled GAG dodecasaccharides using ¹H NMR spectroscopy (Fig. S7), (c) GPC3 from the cell surface was collected and specifically checked using phospholipase C (Fig. 2b), (d) the Fig. S3 was added to help the readers to more clearly understand the varied nomenclature for various labeled GAG saccharides, and (e) quantification of conjugating efficiency and fluorescence intensity were added in the new manuscript (Tab. S2, and Figs. 2d, 4e and 5e), etc. Details can be seen from our revised manuscript and the responses to the reviewers’ comments.

Furthermore, the language of this manuscript has been edited by a professional team from Spring Nature Author Service. Other revisions were also made in the corresponding sections suggested by reviewers. Our responses to all the reviewers’ concerns have been described one by one below. In addition, we have provided a file of the revised version of the manuscript together with another file where the corrections and changes are marked in red for reviewing.

We hope that the manuscript is now acceptable for publication in *Nature Communications*. We thank you for your time and kind concern with our review.

Our responses to the reviewers’ comments have been described below one by one referring to the pages and locations of the marked manuscript in red.

Reviewers' comments:

Reviewer #1, an expert in glycobiology and chemical biology (Remarks to the Author):

This is an interesting manuscript describing novel methodology for addition of GAG oligosaccharides and chains to proteoglycan core proteins, via engineering of sites within the core protein sequence for addition of an aldehyde-bearing formylglycine residue. This allows covalent attachment of oligosaccharides or chains reducing-end labelled with a hydrazide or other related groups which will readily react with the aldehyde under mild conditions compatible with living cells. This permits production of PG core protein labelled with specific GAG moieties, and subsequent evaluation of cell functions mediated by the GAG moieties attached to the core proteins. The authors exemplify the approach by focussing on glypican 3 (GPC3), and its effects on Wnt and Hedgehog signaling. The engineered constructs behaved in the expected manner based on previous data for these interactions. They also demonstrate an additional approach for assessing interactions of PGs with targets which are glycoproteins, through metabolic labelling of glycans with an azido function, that can be tested for coupling with alkyne-labelled GAG moieties on engineered PG core proteins.

This was exemplified by studying azido-labelled Frizzled receptor interaction with alkyne-labelled GPC3 GAG chains.

Overall these are valuable new methodologies which make an important contribution, and open up a lot of new and interesting experimental possibilities for understanding the biology of PGs, and in particular the structure-activity relationships in a true cellular context. However, there are a number of issues with the manuscript in its current form which need to be addressed before I would recommend publication.

Major issues:

1. There are difficulties in understanding the experiments due to poor descriptions and varied nomenclature for different reagents (eg. HaGAG, bioHaGAG, bioHS, HaHS, HS-bio, alkyneHaHS etc). It would be a good idea to add a panel in a figure showing schematically what all these different constructs are, to help the reader follow the experimental rationale. In general the authors need to carefully check their use of terminology in the Results and figure legends, to improve clarity/accuracy.

Our response

Thanks for your good suggestion. To help the readers to more clearly understand the varied nomenclature for various labeled GAG saccharides, we have added a new figure (Fig. S3) that includes a diagram of the structures of the different reagents and their nomenclature. In addition, we corrected some of the abbreviations used to identify the reagents making sure that they are consistent along the manuscript.

2. Results, p5 and Fig 1: the decreasing efficiency of conjugation of oligosaccharides, and the difficulty in getting 2 oligosaccharides attached, is a concern. The authors should provide some information on efficiency of intact chain attachment – is it even lower than the figures for DP15 saccharides? In general this raises a question about whether the statement that “oligosaccharides can be efficiently linked to the aldehyde-tagged core protein” is accurate.

Our response

Thank you for your concerns. Based on your suggestions, we first determined the conjugation efficiency of various size-defined oligosaccharides to GPC3 core proteins by measuring the relative intensity of the corresponding bands in the WB (Fig. 1c). As shown in Table. S2, the conjugation efficiency gradually decreases with the length increase of sugar chains, in particular the case of two chains conjugate to the core protein together, as we mentioned in the text. Regarding the conjugation efficiency of intact chains or longer saccharides (>14 DP) to core proteins, we compared the efficiency of a 5 KD HS fraction (about 18 DP) with HS DP6, and DP12 to AhGPC3-O by WB analysis. As shown in Fig. S10, the conjugation efficiency of the 5 KD HS fraction significantly decreases compared to HS DP12, especially the form with two chains. These results indicate that the increasing charge repulsion and steric hindrance of longer GAG chains can negatively affect the conjugation efficiency.

3. Results, p6: the authors contend that the amount of unmodified core protein is higher in the experiments in living cells, and may be due to cytoplasmic PG not accessible to the labelled oligosaccharides. This question should be checked by specifically analyzing cell surface PGs

stripped from the cell surface for example using proteases.

Our response

It is an important question. We used phospholipase C to release GPC3 from the cell surface for WB analysis. As shown in the new Fig. 2b, the proportion of the glycanated form of GPC3 was significantly increased in the cell surface, and the labeling efficiency of GPC3 on cell surface is about 62%, indicating that the conjugation reaction on living cell surface is more efficient than that in the cell lysate. This result proves our speculation that the increased amount of unmodified core proteins in the lysate of living cells is due to cytoplasmic GPC3 not accessible to the labelled HS oligosaccharide.

4.Results p6: the authors describe using biotinylated HaGAG oligosaccharides (bioHaGAG), but there is no clear description of how these are made. The Methods describes an approach for double labelling GAG chains with hydrazide and biotin, but this is inadequately described and confusing, and its not clear how dual labelled constructs are obtained (reducing end and also intra-chain label ??) and raises questions about the data in Fig S5 and the usage of these constructs. The validation of these dual-labelled constructs needs to be added.

Our response

Thank you for your good suggestion. We have revised the description of the methodology in order to more clearly describe how the double labelling of the GAG chains with hydrazide and biotin was performed (see the corresponding section of “Methods”). Moreover, we have added Fig. S3, which provides additional description of the preparation and chemical structure of the doubly labeled GAG chains. As shown in Fig. S3, the GAG chains were randomly labeled intra chains with biotin, then were partially digested to prepare oligosaccharides, and the biotinylated oligosaccharides were further labeled with hydrazide at reducing ends.

In addition, a binding assay was performed to verify the dual-labeling of HS oligosaccharides as described under “Supplementary information”. In brief, the dually labeled oligosaccharides were conjugated to GPC3-O, and then the conjugated GPC3 was captured by a GPC3 antibody immobilized on well surfaces, and the biotin or alkyne group on HS chains of the engineered GPC3 was detected by TRITC-streptavidin or Cyanine3 azide, respectively. As shown in Fig. S8 in “Supplementary information”, the engineered GPC3 with biotin- or alkyne-labelled HS oligosaccharides showed much higher fluorescence intensity than controls, confirming the dual labeling of HS oligosaccharides with hydrazide and biotin or alkyne.

5.Results, p13: experiments with azido-tagged Frizzled – can the authors rule out the possibility that the azido-tagged Frizzled reacts directly with aldehyde-tagged GPC3?

Fig 5, panel A, the schematic needs to better describe that dual-tagged GAG (with hydrazide and alkyne tags) is needed, and also show the Frizzled as being azido-tagged.

Our response

Thank you for your concern. To investigate whether the azido-tagged Frizzled reacts directly with aldehyde-tagged GPC3, 293T cells were transfected with FZD-7, GPC3-O and hFGE vectors, metabolically labeled with GlcNAz, incubated with HaHS without the alkyne group, followed by click chemical reaction. As shown in Fig. 6c lane 3, no band corresponding to the complex of GPC3-FZD-7 was detected, which rules out the possibility that the azido-tagged Frizzled reacts directly with aldehyde-tagged GPC3.

We revised the Fig. 5a as suggested.

6.The approaches described require complex multiple co-transfections of cells (as many as 4 constructs), but no information is provided about the % transfection efficiencies, or how differential transfection will affect the outcome of experimental usage of the resulting mixed cell cultures. Indeed, the immunofluorescence data seem to show such variation.

Our response

This is a good question. As previously described (1,2), all the signaling results were normalized for transfection efficiency by including in the transfection mixture a beta-galactosidase expression vector. The results of the luciferase activity were then normalized by the corresponding beta-galactosidase activity.

Reference

1. Capurro, M.I., Xiang, Y.Y., Lobe, C. & Filmus, J. Glypican-3 promotes the growth of hepatocellular carcinoma by stimulating canonical Wnt signaling. *Cancer research* 65, 6245-6254 (2005).
2. Capurro, M.I. et al. Glypican-3 inhibits Hedgehog signaling during development by competing with patched for Hedgehog binding. *Developmental cell* 14, 700-711 (2008).

7.Some of the background experiments to validate hydrazide labelling are insufficient and not well described. From Fig S4 it seems that the only measure for % labelling for larger saccharides is claimed resistance of the reducing terminal tetrasaccharide to cleavage by heparin lyases. More information by other methods is needed to validate the labelling.

Fig S4 is also missing information on panel C and the MS data is not convincing (shows just a series of spikes at high m/z and no information on background noise).

Our response

Thanks for your concerns and suggestions. The structures of the larger GAG oligosaccharides are extremely diverse and complex, and thus the labeled and non-labeled large GAG chains cannot be effectively separated and quantified by HPLC. Fortunately, we found that the hydrazide labeling completely inhibits the cleavage of the reducing-end tetrasaccharides of GAG chains by most GAG lyases, and thus the labeling efficiency of oligosaccharides could be determined by calculating the ratio of the labeled tetrasaccharides to disaccharides at the reducing end in the digest of hydrazide-labeled GAG saccharide chains. Relevant background was added in the section of "Preparation of hydrazide-labeled GAG oligosaccharides" in "Supplementary information".

To further validate the labeling, hydrazide-labeled GAG oligosaccharides (DP12) were analyzed by ¹H NMR spectroscopy. As shown in Fig.S7, compared with the non-labeled samples, the labeled GAG oligosaccharides have some weak but significant signals at 2.20-2.27ppm and 1.55-1.65ppm specifically corresponding to ¹H of CH₂ structures in the labeling molecule (adipic dihydrazide), as we discussed in the text.

In addition, the labeling tetrasaccharides from the reducing ends of hydrazide-labeled HS decasaccharides were analyzed by MS again and the new result was shown in the new Fig. S6d.

Minor issues:

1.Abstract: the authors should be careful to indicate that both GAG oligosaccharides and GAG

chains can be attached using the methodology. In fact most (but not all) of the data is achieved with oligosaccharides. In general in the text the authors need to be more clear in describing oligosaccharides vs chains in the various experiments.

Our response

Sorry for the ambiguous statement. We have revised these unclear statements in the corresponding sections of the manuscript.

2.Results, p6: reference to Fig S4 should be S5?

Our response

Sorry for this mistake. We have corrected it.

3.Results, p8: the details of how DEAE chromatography was used needs to be added. Use of the term HS-Bio needs to be better defined. For Fig 3a the text says GAG chains, but in fact oligosaccharides are used.

Our response

Thank you very much. The experimental procedure to concentrate GPC3 with DEAE-Sepharose has been added to the methods section. Biotinylated HS is abbreviated as BioHS in the revised manuscript. We have corrected the unclear statement in the legend of Fig. 3a.

4.Fig 3, legend: panel A need to define better which oligosaccharides and “chains” are used. Fr 1 and 2 need to be properly defined and their source and preparation given in the Methods.

Our response

Thanks. We have revised the legend. The HS Fr 1 and Fr2 have been properly defined and their source and preparation were also described in the Methods Section (see “Preparation of hydrazide-tagged GAG oligosaccharides”). Fr.2 is the oligosaccharide fraction isolated from the digest of HS with Heparinase III, which is longer than DP14 but shorter than Fr.1 (Fig. S4); Fr.1 is the oligosaccharide fraction longer than Fr.2. In addition, the sources of the GAG polysaccharides: HA, CS, DS, HS and Hep are described in “Materials”.

5.Fig 4, legend: panel A, the use of HS-Bio is confusing since the authors seem to be using TRITC-streptavidin to detect biotinylated HS.

Our response

Thanks. We have changed HS-Bio to BioHS that represents biotinylated HS.

6.Fig S7, legend: what concentrations of saccharides are used in panels b and c ? Fr1 and Fr2 saccharides are not defined.

Our response

Thanks. As shown in the legend, the final concentration of saccharides used in panels b and c was 1 μ M. The HS Fr 1 and Fr2 have been defined in the section of “Preparation of hydrazide-tagged GAG oligosaccharides”, as mentioned above.

7.There are large numbers of typos and poor grammar usage which need to be corrected to improve the clarity of the writing.

Our response

Thanks for your suggestion. We have checked and corrected the grammar and spelling mistakes in the full text. In addition, the language of this manuscript has been edited by a professional team from Springer Nature Author Service.

Reviewer #2, an expert in proteoglycan engineering (Remarks to the Author):

The manuscript by Wang and coworkers describes a method for generating semi-synthetic proteoglycan structures directly at the surface of live cells. Proteoglycans are key regulators of growth factor signaling activity in cells linked to their proliferation and differentiation. They are of great interest as targets to manipulate cellular activity but have been very difficult to study due to the complexity and non-templated biosynthesis of their glycosaminoglycan (GAG) modifications. Many efforts have been developed to gain control over the GAG component of cellular signaling, some are cited in this paper; however, the ability to control GAG glycosylation patterns of individual proteoglycans on the cell surface has not been achieved. In this context, the present work constitutes a significant step forward in this area. The authors approach the problem of controlling PG glycosylation by taking advantage of the aldehyde tag technology developed by Bertozzi. By replacing the native glycosylation sites in the proteoglycan, glypican 3 (GPC3), with short sequences containing cysteine, which can be converted into formyl glycine residues by the action of the formyl glycine modifying enzyme, the authors introduced new sites for chemical conjugation of hydrazide modified HS chains with well-defined structures directly at the cell surface. They then proceed to evaluate the activity of their neoglycoconjugates in signaling through Wnt and SHh proteins and use a combination of metabolic oligosaccharide engineering and click chemistries to capture the signaling complexes formed in the presence of the engineered PG structures. This study is very promising with potentially high impact on the field; however, there are some significant experimental issues and problems with data interpretation, due to lack of appropriate controls, that raise questions about the conclusions of this study. While my concerns regarding the described experiments and conclusions are described in detail below, the most critical issues the authors need to address prior to publication, in my opinion, are the harsh conditions (prolonged exposures of cells to low pH environments) needed to construct the neoglycoconjugates on the surfaces of living cells in the absence of assessment of cell viability and effects on cell function, the inefficiency (or lack of quantification) of the conjugation process, and the lack of important controls in the signaling and crosslinking assays that bring the validity of the conclusions into question. Following are my specific comments regarding the different aspects of this manuscript:

1. Generation of hydrazide-labeled GAGs. The manuscript text or the methods section (page 23) does not specify the pH of the buffer used for HS labeling with adipic hydrazide. The efficiency of GAG functionalization was determined either by purification of the conjugates for short disaccharides or by Heparase III digest in combination with MS analysis for longer oligosaccharides. The compositions of the oligosaccharides after digest were confirmed by disaccharide analysis. I wasn't able to find the SEC traces for the purified disaccharide conjugates, only the crude mixtures before purification were presented. The traces for the purified GAGs should be included in the SI. It is a bit challenging to follow all the different GAGs generated and used in the study. The authors should include a table in the SI summarizing all of the structures, their disaccharide composition, and the yields for hydrazide conjugation.

Our response

Thanks for your suggestion. At the beginning of the study, we optimized the reaction buffer for HS labeling with adipic hydrazide. The results showed that the labeling efficiency was highest at a pH of 5.0. We have now specified the pH of the buffer in the revised manuscript. In addition, the SEC

traces of the purified hydrazide-labeled disaccharide and hydrazide-labeled tetrasaccharide conjugates were added in Fig. S5 and S6.

We have added a new figure (Fig. S3) to more clearly show the structures and nomenclature of the labeled GAG oligosaccharides, and added a new table (Tab. S1) to show the disaccharide composition of the GAG polysaccharides used in the study. The yields of hydrazide conjugation for the various GAG oligosaccharides were added to the text in the “Supplementary information”.

2. Generation of GPC3 mutants. The text on page 4 describing the generation of mutants and data in Fig 1 and S1 do not specify what cells these mutants were expressed in. Is it HEK293T cells as described later in the text?

Our response

Thanks a lot. Yes, these GPC3 mutants were expressed in 293T cells. In the section of “FGly conversion and verification in GPC3 mutants” in Methods, we have described the generation of GPC3 mutants in detail. We have specified the cells used in the section of “Results” section too.

I assume HEK293T cells don't express endogenous GPC3, can the authors confirm that? This should be clarified earlier in text and in Figures 1 and S1.

Our response

HEK293T cells don't express GPC3 (1,2), we have clarified this in the text.

References:

1. Capurro, M.I., Xiang, Y.Y., Lobe, C. & Filmus, J. Glypican-3 promotes the growth of hepatocellular carcinoma by stimulating canonical Wnt signaling. *Cancer research* 65, 6245-6254 (2005).
2. Capurro, M.I. et al. Glypican-3 inhibits Hedgehog signaling during development by competing with patched for Hedgehog binding. *Developmental cell* 14, 700-711 (2008).

Expanded western blots in Fig 1b/c (full length) for all mutants and wild type GPC3 should be included in SI and it should be clarified what GPC3 is. Is this wild type fully glycosylated GPC3 expressed by the HEK293T cells?

Our response

Thanks for your suggestion. We consider that the WB results in Fig 1b/c are the basis of the following studies, which display the wild-type fully glycanated GPC3, and the non-glycanated mutants GPC3 Δ GAG, GPC3-S495C and GPC3-O. Thus, we keep this data in the text rather than SI.

Yes, the fully glycosylated GPC3 was expressed by the HEK293T cells.

What was the efficiency of conversion of Cys to fGly? Some data are presented in Fig S1 but it is unclear how efficient the process is overall and particularly for GPC3-O. MS data should be included.

Our response

Thanks for your good suggestions. To investigate the efficiency of conversion of Cys to fGly, 293T cells were co-transfected with GPC3 and hFGE vectors, and the GPC3-O protein in the cell lysate was purified by affinity chromatography using the GPC3 antibody followed by carbamidomethylation with 2-iodoacetamide and digestion with trypsin. The tryptic fragments of

GPC3-O were analyzed on a linear ion trap-Orbitrap hybrid mass spectrometer coupled with a nano-LC system as described in “Supplementary Note 1: Confirmation of the Cys-to-fGly conversion in GPC3 mutants”. The results are shown in Fig. S2. By analyzing the integral area of corresponding peptide peaks, the efficiency of conversion Cys to fGly at each mutant site of GPC3-O is about 90%, and about 78% for both sites, as described in the revised manuscript.

3. In vitro GAG conjugation to GPC3 mutants. The authors state that the double mutant GPC3-O was conjugated with one and two chains in a ratio of 50% to 20% due to close proximity of the two aldehyde tags, with 30% of the protein remaining unconjugated. It is possible, but this can also be caused by incomplete Cys-to-fGly conversion in the aldehyde tag. In general, the conjugation of hydrazide-labeled HS to the mutants seems to proceed, but the question is what the overall efficiency of the process is. Quantitative analysis of the Cys-to-fGly conversion by MS would confirm the authors' conclusion regarding steric effects on would give clear indication as to the efficiency of the process.

Our response

Thanks. As described in our answer to the previous question, the efficiency of Cys-to-fGly conversion on GPC3-O was determined by MS. The efficiency was calculated by measuring the integral area of the corresponding peptide peaks. It can be inferred from the results that about 78% double mutant GPC3-O was converted, which is much higher than the conjugation efficiency with two chains. In addition, the efficiency of GPC3-O conjugation with two chains decreases with the length increasing of GAG chains (Table S2), which is consistent with our proposal that the charge repulsion and steric hindrance between GAG chains have a detrimental effect on the conjugation efficiency.

In Figure 1b/c, does the first GPC3 band correspond to fully glycosylated wild type GPC3 expressed in the HEK293T cells? Or is it one of the mutants treated with an oligosaccharide mixture? This should be clarified and if the latter is the case, a fully glycosylated wt GPC3 control expressed in HEK293T cells should be included for comparison.

Our response

Thanks. As we explained above, the first GPC3 band correspond to fully glycosylated wild type GPC3 expressed in the HEK293T cells as a control. We have revised the name of GPC3 to wt GPC3 in the related figures and annotated it in the legends.

The authors also generated the Cys509 mutant, but HS conjugation data is not provided in Fig 1 or in SI. The Cys509 mutant was used in cell signaling assays in this study but data for hydrazide conjugation to this protein in vitro or on cell surfaces was not provided. Is conjugation to this site as efficient as to fGly 459? Is the yield of conversion of Cys509 to fGly509 the same for Cys459 to fGly459? Could differences in fGly generation at these two sites explain why the doubly HS modified GPC3-O forms less efficiently?

Our response

Thank you for your concerns. As shown in Fig. S1, the conversion efficiencies of Cys509 to fGly509 and Cys459 to fGly459 are similar. Furthermore, as we found in the MS analysis, the two sites in the dual mutant GPC3-O have similar conversion yields (approximately 90%). In addition, we have determined the conjugation efficiencies of various oligosaccharides in the three mutants (GPC3-

S495C, GPC3-S509C and GPC3-O) based on WB assays as described in the revised manuscript. As shown in Table S2, the reduction in conjugation efficiency of GPC3-O with increasing GAG lengths was less pronounced than that observed with GPC3-O displaying two chains. These results also strongly support the hypothesis that the reduced conjugation efficiency of two chains to GPC3-O is due to charge repulsion and steric hindrance between highly negatively charged GAG chains in particular long chains.

4. Cell surface HS conjugation to GPC3 mutants. The hydrazide conjugation reaction requires low pH to proceed and, accordingly, the authors subjected the cells for prolonged periods of time 1-8 hrs to pH 6 conditions. An obvious and significant concern is the damage to the cells under those conditions. No data is provided regarding the cells' viability and proliferation capacity after the treatment. Since the authors evaluate sensitive signaling responses after such a treatment, these assays with appropriate controls should be included.

Our response

Thanks for your suggestions. The effect of conjugation reaction on the viability of cells has been investigated using the MTT method, as described in the revised manuscript. As shown in Fig. S9c, at pH 6 the viability of the cells is reduced to 72% at 8 hrs, which is acceptable for subsequent experiments. To reduce cell damage, we used a 4 hrs-treatment. At this time point viability is only reduced by 15 %. After conjugation, the buffer is replaced with normal medium and therefore we can conclude that viability should not affect the results of the signaling experiments performed subsequently (Fig. S9c).

The authors should quantify the extent of HS conjugation to the GPC3 mutants at the cell surface. The western blot in Fig 2B shows that the process is clearly inefficient and concentration of the conjugated proteins by ion exchange was needed to for the conjugates to clearly manifest on the blot. Perhaps this can be addressed by using a different conjugation method not requiring low pH conditions.

Our response

Thanks a lot for your concerns. As we mentioned in the manuscript, the seemingly low conjugation efficiency on cell surface could be due to the inaccessibility of HaHS to the GPC3 in the cytoplasm of cell. To confirm this possibility, we have used phospholipase C to release and collect GPC3 from the cell membrane to be able to measure the labeling efficiency of the GPC3 on the cell surface. As shown in new Fig. 2b in the revised manuscript, about 62% of GPC3 on cell surface was labeled with HS oligosaccharides, suggesting that the labeling of GPC3 on living cell surface is efficient as that in cell lysate.

In addition, we are trying to find other method that could be used to efficiently conjugate GAG chains to PG core proteins to produce more stable engineered PGs under neutral conditions, but frankly it is not easy work.

Page 6, line 3 states that the HS GAGs were conjugated to the transfected cells under optimized conditions in SI Fig S4. This figure shows SEC analysis of hydrazide HS. The text should specify what the optimize conditions are (presumably, 1 uM HS, pH 6 for 6 hrs at 37 deg C??).

Our response

Thanks for the suggestion. We have specified the optimal conditions (1 μ M HS, pH 6 for 4h at 37

°C) in the text of the revised manuscript. Sorry for the mistake, it should have been FigS5 instead of FigS4. Now it is FigS9 since we added some new figures.

Fig S5c describes the surface residence time of the conjugates (labeled as “vanishing time”). Is the loss of surface HS due to endocytosis by the cells or hydrolysis of the hydrazide linker? What conditions (ie, pH) is this experiment performed in? Assessment of the stability of the linker, which is prone to hydrolysis, should be provided.

Our response

Thanks for concerns. As reported previously, aldehydes react with hydrazines to form hydrazones, which are relatively stable under physiological conditions (1, 2). Nevertheless, they could be gradually hydrolyzed (3,4). We think that the disappearance of the engineered GPC3 with specific GAG chains from the cell surface could be caused by several factors including the hydrolysis of the hydrazone linker, digestion by some endogenous GAG- or/and protein-degrading enzymes and endocytosis by the cells, as mentioned in the text in Supplementary information.

Reference

1. Wu, P. et al. Site-specific chemical modification of recombinant proteins produced in mammalian cells by using the genetically encoded aldehyde tag. *Proceedings of the National Academy of Sciences of the United States of America* 106, 3000-3005 (2009).
2. Bioconjugation with Aminoalkylhydrazine for Efficient Mass Spectrometry-Based Detection of Small Carbonyl Compounds *ACS Omega*. 4, 13447–13453 (2019).
3. Agarwal, P. et al. Hydrazino-Pictet-Spengler Ligation as a Biocompatible Method for the Generation of Stable Protein Conjugates. *Bioconjugate Chemistry* 24, 846–851 (2013)
4. Wendeler, M. et al. Enhanced Catalysis of Oxime-Based Bioconjugations by Substituted Anilines. *Bioconjugate Chemistry* 25, 93–101 (2014)

Fig 2: the fluorescence micrographs in the figure are rather small and hard to evaluate visually. The authors should consider adding to the SI quantitative analysis for immunofluorescence and FRET efficiency as well as higher resolution images.

The use of biotinylated HS chains to monitor conjugation to the GPC3 mutants on the cell surface provides evidence for this process proceeding as described; however, it's difficult to tell from the fluorescence micrograph in Fig 2c what the background binding of hydrazide-HS is to the surface of control cells treated with the empty vector. Those controls were not included in the western blots in Fig 2b. A more sensitive and quantitative assessment of conjugation using flow cytometry might be beneficial here. Also, probing for sufficient HS conjugation more directly by using standard HS-binding proteins such as FGF2 or the anti-HS antibody 10E4 instead of relying on biotin-streptavidin interaction may be beneficial. Although endogenous HS on the cell surface may pose a challenge in such an experiment.

Our response

Thanks for your suggestions. The resolution of the images has been improved, the immunofluorescence intensities of corresponding cells have been quantitatively estimated by flow cytometry (Fig. 2d), and the FRET efficiency (38%) between the FITC-stained core protein and the TRITC-stained side chain has been calculated and added to the text too. As shown in Fig. 2d, the fluorescence intensity of cells with GPC3 bound biotinylated HS chains is much higher than those of various negative controls. Moreover, we have tried to use anti-HS/Hep antibody T320.11 instead

of streptavidin to detect HS conjugation, but the background signal is too high to distinguish the test cells from the control cells due to the large amount of endogenous HS on the cell surface, as you speculated. To reduce the interference of endogenous HS, we treated the cells with heparinases before adding the HaHS. As shown in the following Figure, the background could be significantly reduced by the treatment with Heparinases, thereby highlighting the signal of HS chains conjugated to GPC3 core proteins on cell surface, further indicating that aldehyde-labeled GPC3 on the cell surface could be conjugated with HaHS.

Figure: **Immunofluorescence detection of the GPC3 core protein and the attached HS chains.** Cells were transfected with GPC3-O and hFGE vectors. Two days after transfection, GPC3-O expressing cells were treated with or without Heparinase I II III (hepase)(10mU) for one hour before incubation with HaHS. The GPC3-O core protein and HS chains were detected with Anti-Heparin/Heparan Sulfate Antibody clone T320.11 and FITC-conjugated goat anti-mouse secondary antibody, and Rabbit Anti-Glypican 3 antibody and TRITC-conjugated goat anti-rabbit secondary antibody, respectively.

5. *Signaling response in GAG-engineered cells. The observations of increasing Wnt activity in response to increasing HS length is interesting. The authors should comment on the influence of conjugation efficiency on signaling activity, since they indicated that the reaction yield decreases with increasing HS chain size. This may mean that high levels of glycosylation may decrease Wnt activity. In these experiments, soluble HS lacking the hydrazide unit should be included. The role of HS in Wnt signaling can be complex and is often poorly defined. For instance, cell surface HS can inhibit Wnt signaling by sequestering the protein away from its receptors, while soluble heparin has been shown to promote Wnt association with cognate receptors and activate signaling. It appears that the signaling assay was performed without washing the excess soluble hydrazide-HS. It is not possible to tell from these experiments if the changes in signaling through Wnt (or Shh) in Figure 3 (and 4) were the result of activity from cell-surface or the unwashed soluble HS chains.*

Our response

Thank you for your detailed question. We have now tested the effect of free HS on the GPC3-induced stimulation of Wnt/Shh signaling, unlabeled various GAG PD14 oligosaccharides have

been used as controls in signaling experiments. As shown in Fig. S12, GPC3-O-transfected cells treated with hydrazide-unlabeled GAG oligosaccharide chains, which cannot be conjugated to the GPC3 core protein, have no significant impact in the activation of Wnt and Shh signaling. The results clearly show that the conjugation of Hep/HS chains to the GPC3 core protein on cell surface is required for the effective regulation of Wnt and Shh signaling.

Previous studies have shown that GPC3 Δ GAG without glycanation has lower Wnt3A and Hh-stimulating activity than wild type GPC3 (1,2), which is consistent with our results showing that the non-glycanated GPC3-O shows relatively weak activity compared with the wt GPC3. This indicates that the increasing Wnt and Hh-stimulatory activity in response to the increasing length of the oligosaccharides is probably underestimated due to the decreasing conjugation efficiency.

Reference:

1. Capurro, M.I., Xiang, Y.Y., Lobe, C. & Filmus, J. Glypican-3 promotes the growth of hepatocellular carcinoma by stimulating canonical Wnt signaling. *Cancer research* 65, 6245-6254 (2005).
2. Capurro, M.I. et al. Glypican-3 inhibits Hedgehog signaling during development by competing with patched for Hedgehog binding. *Developmental cell* 14, 700-711 (2008).

The authors should also clarify whether GPC3 is the fully glycosylated wild type protein? In most of the figures it's referred to as "GPC3 core protein" but it is not clear if it is glycosylated. The presence of endogenous HS on the surface of the HEK291T cells and its possible effects on growth factor binding and activity should be discussed in the manuscript.

Our response

Sorry for the ambiguous statement. As we mentioned above, we have revised GPC3 to wt GPC3 for wild type GPC3 with full glycanation in the corresponding Figures based on your suggestion. At the same time, the effects of endogenous cell surface HS on growth factor binding and activity have been discussed in the text.

The harsh acidic treatment of cells for prolonged periods of time is of concern. The authors should clarify if the included controls were subjected to the same treatment and how that affected signaling responses with respect to cells treated under physiological conditions.

Our response

Thank you for your concern. As we described in "Methods" section, the cells of the control group were treated under the same reaction conditions (pH 6.0) as those from the test groups. After conjugation the reaction buffer was immediately replaced with normal cell culture medium to perform the cell signaling assays under physiological conditions. In addition, as mentioned above, the harm of the acidic treatment is small and acceptable (Fig. S9d), under the optimized reaction conditions (1 μ M HS, pH 6 for 4h). This is confirmed by the fact that the cells responded to Wnt and Hh treatment as expected.

The authors indicate, based on fluorescence immunostaining data in Fig 3, that Wnt3a binds to GPC3 in the absence of HS modifications. Did the authors consider that Wnt3a might bind to endogenous HS proteoglycans on the cell surface besides GPC3, which would be expected in cells with fully functional HS biosynthesis? What is the evidence that Wnt3a binds to the non-glycosylated GPC3 and not another endogenous HSPG on the cell surface?

Our response

This is good question. Our previous studies have shown that Wnt3a binds to non-glycanated GPC3 (GPC3 Δ GAG) with high affinity(1). In addition, we have shown that in our conditions the binding of Wnt3a to 293T cells that have not been transfected with GPC3 is undetectable (1, 2). Thus, we believe that Wnt3a mainly binds to the non- glycanated GPC3 rather than endogenous HSPG on the cell surface.

Reference:

1. Capurro, M.I., Xiang, Y.Y., Lobe, C. & Filmus, J. Glypican-3 promotes the growth of hepatocellular carcinoma by stimulating canonical Wnt signaling. *Cancer research* 65, 6245-6254 (2005).
2. Li, N. et al. A Frizzled-Like Cysteine-Rich Domain in Glypican-3 Mediates Wnt Binding and Regulates Hepatocellular Carcinoma Tumor Growth in Mice. *Hepatology*, 70, 1231-1245 (2019).

The SHh signaling experiments suffer from the same technical issue as the Wnt signaling experiments, which is the presence of soluble unreacted hydrazide-HS during the course of the assay. Soluble HS can inhibit SHh signaling and may be responsible for the observed inhibitory activity (including dose dependence). The authors need to demonstrate that it is not the soluble or otherwise adsorbed HS that is responsible for the observed changes in signaling (negative controls consisting of cells treated under identical conditions with HS lacking the hydrazide functionality should be included).

Our response

Thank you for your concern and suggestion. As we mentioned above, unlabeled various GAG oligosaccharides lacking the hydrazide group have been used as controls in signaling the experiment. As the results shown in Fig S12b, the treatment with hydrazide-unlabeled GAG oligosaccharides has no significant effect on the Shh signaling of GPC3-O-transfected cells, just like the case of Wnt signaling.

The final part of the paper describing the capture of GPC3 with alkyne-modified HS and FZD-7 with azide-modified N-glycans through proximity induced cycloaddition reaction in the absence of copper catalyst is not convincing. Proximity induced alkyne-azide cycloaddition, while demonstrated early on by Sharpless in catalytic sites of enzymes, is rarely observed in the case of relatively weak interactions, such as those between GAGs and GF/receptor complexes. The included controls using cells expressing only one of the protein components do not provide strong enough evidence that covalent crosslinking is indeed occurring in this system. Controls performed with metabolically labeled FZD-7 and GPC3 modified with HS without alkynes were missing in Fig 5. What is the rationale for the core or LacNAc extensions of the N-linked glycans to be in close proximity to the HS chain? What is the efficiency of GlcNAz labeling of N-glycans in FZD-7? In general, GlcNAz incorporation into N-glycans is not as effective as the incorporation of GalNAz, which undergoes efficient epimerization to produce UDP-GlcNAz.

Our response

Thank you for your concerns and suggestions. As we described in the section of “Interaction assay for GPC3 and target protein FZD-7” in “Methods”, the click reaction was performed as follows: “the cells were incubated with PBS containing 250 mM tris-(benzyltriazolylmethyl)amine (TBTA), 50

mM CuSO₄, and 500 mM L-ascorbic acid sodium salt (SA) for 2 h at room temperature”, in which the 50 mM CuSO₄ was used as the catalyst. Notably, in this stage of the experiment the harm of Cu²⁺ to cells no longer needs to be considered.

According to your suggestion, a control performed with metabolically labeled FZD-7 and GPC3 modified with HS without alkynes has been added to Fig. 5c lane 3. Briefly, HEK293T cells were transfected with FZD-7 and GPC3-O vectors, metabolically labeled with GlcNAz, incubated with HaHS DP14 without alkyne group, and then the click chemical reaction was performed. The result shows that only one band corresponding to unconjugated FZD-7 but no bands corresponding to the complex of FZD-7-GPC3 are detected.

Based on your suggestion, we have investigated the labeling efficiencies of labeled FZD-7 with GalNAz and GlcNAz, respectively. Briefly, HEK 293T cells were transfected with FZD-7, and incubated with different concentrations of GalNAz or GlcNAz. The cell lysates were added to the wells covered with FZD-7 antibody, and after incubation and blocking the metabolically labeled FZD-7 was detected using Cy3 Alkyne through a click reaction. As shown in the Figure below, the labeling efficiency of GalNAz is indeed higher than that of GlcNAz, but the difference is very small. Notably, in the interaction assay we usually try to reduce the labeling efficiency of the sugar chains in order to avoid the adverse effects of over-labeling on the natural function of glycoproteins, because the excessive introduction of unnatural components will change the natural structure of the sugar chains too much.

Regarding the rationale for the core or LacNAc extensions of the N-linked glycans to be in close proximity to the HS chain, as we mentioned in the manuscript, previous studies showed that GPC3 can interact with the Wnt receptors frizzleds (FZDs) including FZD-7 through its GAG chains, indicating that the N-linked glycans of FZD-7 should be close enough to the HS chains of GPC3. Our results are consistent with our previous findings.

Figure FZD-7 Metabolic labeling of FZD-7 with GlcNAz or GalNAz.

Reviewer #3, an expert in proteoglycan signalling (Remarks to the Author):

In this study, using glypican-3 as a model, the authors applied an aldehyde tag-based approach to assemble PGs with GAG chains, including different types GAG and different sized HS. Functional studies showed that the engineered glypican-3 can regulate Wnt and Hedgehog signaling like the wild type, and the regulation mainly is mediated by assembled HS, not by other GAGs. More HS focused study further showed that the regulatory activity of engineered glypican-3 on Wnt and Hedgehog signaling depends on size and chain number of assembled HS. In combination with metabolic glycan labeling, this approach was extended to study and confirmed that glypican 3 interacts with Frizzled 7. In summary, this aldehyde tag-based approach has been clearly shown to be applicable for the structural and functional studies of proteoglycans with specific GAG chains.

Major concern:

1. Technically not novel: The aldehyde tag-based assembly of carbohydrate moiety to glycoprotein and metabolic glycan labeling both have been well established and widely used in glycobiology. This study applies this approach to study proteoglycan.

Our response

Thank you for your comments. We agree that the aldehyde-tag and metabolic glycan labeling technologies used in this study are not new, and have been successfully used in the glycobiology field, as we mentioned in the manuscript. However, compared with glycoproteins, the structures of proteoglycans are much more complex due to their highly heterogeneous GAG polysaccharide chains, which present a big challenge for the structural and functional studies of proteoglycans. As we mentioned in the manuscript, to date we still have no method to precisely control the structure of GAG chains that are inserted into the core protein of proteoglycans *in vivo*, and thus it is not possible to investigate the effects of GAG side chains with specific structural features on the functions of proteoglycans. In this study, by combining the aldehyde-tag and metabolic labeling techniques we provide a very useful platform for the structural and functional studies of proteoglycans with specific GAG chains.

2. Application study did not result in any new discovery, only confirmed reported findings in literature.

Our response

Thanks for your comment. In this study, there are a series of new discoveries, such as, the effects of different types of GAG side chains on the activity of GPC3 in Wnt or Shh signaling; the minimal size requirement of HS/Hep side chains for the full signaling activity of GPC3; and a demonstration that GPC3 directly interacts with FZD-7 on the cell surface (Our previous studies only provided some indirect evidence for this interaction).

3. Statistical analyses are not clearly stated: Which tests were used ?, how many repeats for each figure ? Except Fig.4C, no p values for all other bar figures are included.

Our response

Thank you for your concerns. In this manuscript, each experiment was repeated at least three times, and in the case of quantitative experiments such as cell signaling assays and binding assays, each experiment was done at least three times by triplicates. As stated in the revised manuscript, Student's

t-test (two-sided) was carried out for comparison of the statistical differences between two groups.

REVIEWERS' COMMENTS

Reviewer #1 (Remarks to the Author):

The authors have adequately addressed my concerns with additional experiments, revisions and comments, and the work is now suitable for publication.